# The North Asian Genus *Kolhymamnicola* Starobogatov and Budnikova 1976 (Gastropoda: Amnicolidae), Its Extended Diagnosis, Distribution, and Taxonomic Relationships [†]

**Tatiana Sitnikova** [1,2,*], **Tatiana Peretolchina** [1], **Larisa Prozorova** [3], **Dmitry Sherbakov** [1], **Eugeny Babushkin** [2,4] **and Maxim Vinarski** [2]

1   Limnological Institute SB RAS, 3 Ulan-Batorskaya Str., 664033 Irkutsk, Russia;
    tatiana.peretolchina@gmail.com (T.P.); sherb@lin.irk.ru (D.S.)
2   Laboratory of Macroecology & Biogeography of Invertebrates, St.-Petersburg State University,
    7/9 Universitetskaya Emb., 199034 St.-Petersburg, Russia; babushkines@gmail.com (E.B.);
    m.vinarsky@spbu.ru (M.V.)
3   Federal Scientific Center of the East Asia Terrestrial Biodiversity, Far Eastern Branch of the Russian Academy
    of Sciences, 159 100-years of Vladivostok, 690002 Vladivostok, Russia; lprozorova@mail.ru
4   Scientific and Educational Center of the Institute of Natural and Technical Sciences, Surgut State University,
    1 Lenin Avenue, 628403 Surgut, Russia
*   Correspondence: sit@lin.irk.ru
†   urn:lsid:zoobank.org:pub:FF391FCB-3C36-4DEB-B7D5-7980FB9AF870.

**Abstract:** The taxonomic position and phylogenetic affinities of the endemic North Asian genus *Kolhymamnicola* Starobogatov and Budnikova, 1976 (Gastropoda: Amnicolidae) remain unknown. To resolve this, we studied key morpho-anatomical characteristics of *Kolhymamnicola* snails and performed a molecular phylogenetic analysis based on sequences of COI mtDNA, 16S rRNA, and 18S rRNA genes. In terms of protoconch microsculpture, operculum, radular teeth, and gill complex morphology, *Kolhymamnicola* snails do not differ significantly from the North American genera *Amnicola* Gould and Haldeman, 1840 and *Taylorconcha* Hershler et al., 1994, and the European genus *Marstoniopsis* van Regteren Altena 1936. The bifid penis found in *Kolhymamnicola* is similar to that in the genus *Marstoniopsis*. The female reproductive anatomy has some features shared by *Kolhymamnicola* and *Taylorconcha* (absence of bursa copulatrix, single seminal receptacle in $rs_{2'}$ position, and ventral channel). The molecular analysis has revealed Taylorconcha as the closest relative to *Kolhymamnicola*; the COI-based genetic distance between them amounted to 0.113. We discuss the possible time of divergence of these two genera, as well as of European *Marstoniopsis* and the Baikal Lake endemic family Baicaliidae. The last common ancestor of these groups was widely distributed in Miocene–Pliocene in the Holarctic waterbodies. Recent *Kolhymamnicola* snails are distributed in Northern Asia, including lakes of the Baikal rift zone. We rank the Baicaliidae as a family rather than a subfamily of Amnicolidae based on their distinct, unique morpho-anatomical characteristics and highly supported separate position on the molecular tree. The tribe Erhaiini Davis and Kuo, 1985 is elevated to the rank of the family, with 3–4 recent genera included. The family Palaeobaicaliidae Sitnikova et Vinarski **fam. nov**. is established to embrace the Cretaceous North Asian gastropods conchologically similar to the recent Baicaliidae and Pyrgulidae.

**Keywords:** truncatelloid snails; protoconchs; radular teeth; reproductive system; morphological comparison; molecular phylogeny



## 1. Introduction

The family Amnicolidae Tryon, 1863, as it is treated in the recent literature [1–3], is divided into two subfamilies, Amnicolinae and Baicaliinae P. Fischer, 1885. The rank of the latter taxon is still controversial since some authors, mostly Russian, consider it a separate

family, sister to the Amnicolidae s. str. [4–7]. The range of Amnicolinae is broadly Holarctic, with 14 recent genera distributed in Europe, North and East Asia, and North America [3]. Both generic composition and phylogenetic relationships of this group have been debated during the last several decades, and, as a consequence, its position in the Caenogastropoda system varied from the status of a subfamily within the family Bithyniidae [8] to the rank of a family including the group of endemic Baikal gastropods known as Baicaliidae [2]. The most recent phylogenetic study [9] identified Amnicolidae s. lato as a sister group to a clade uniting the families Bythinellidae Locard, 1893 and Fontigentidae D.W. Taylor, 1966. However, to date, only 6 of 11 nominal genera included in the subfamily Amnicolinae are studied genetically. The morphological information on some of the amnicolid genera is also limited.

The genus *Kolhymamnicola* Starobogatov and Budnikova, 1976 belongs to this group of understudied amnicolid taxa. Most of the currently available knowledge about this genus is limited to information published over 30 years ago, in the premolecular period of the development of malacological systematics [10–13]. The snails of this genus are relatively rare in North Asian waterbodies and, thus, remain underrepresented even in large malacological collections. The morphological information on *Kolhymamnicola* is scarce, whereas molecular data are unavailable altogether. The first nominal species of this group, *Amnicola miyadii* Habe, 1942, was discovered in ponds on the Kurile archipelago's Shumshu (North Kuriles) and Kunashir (South Kuriles) islands; the former was designated as the type locality [14]. Another North Asian species, *A. kolhymensis* Starobogatov and Streletzkaja, 1967, was described from the floodplain of the Kolhyma River, Yakutia (northeastern Siberia). Starobogatov and Budnikova [11] combined these two species under the generic name *Kolhymamnicola* (with *A. kolhymensis* as the type species), and, later, Starobogatov [15] proposed to erect a new family, Kolhymamnicolidae, to include two genera, *Kolhymamnicola* and *Akiyoshia* Kuroda and Habe, 1954. This opinion was accepted by some subsequent authors [5,13].

In 1988, two more species of *Kolhymamnicola*, *K. ochotica* Zatravkin and Bogatov 1988 and *K. wasiliewae* Zatravkin and Bogatov 1988, with type localities situated in the Okhotsk Sea coast and in the Amur River basin, respectively, were described [12].

Though Kolhymamnicolidae is no longer accepted as a separate family [2], the species composition of *Kohlymamnicola* has not changed since 1988. The genus is now treated as including the four nominal species listed above [7].

*Kolhymamnicola* snails were found in macrophyte thickets or on the bottom substrate of the shallow water zone of lakes in various regions of Northern Asia: The Kolhyma River basin [10,16], Chukotka [11], Kamchatka peninsula [11,17–19], the basin of the Lower Lena River [20], waterbodies of the Okhotsk Sea coast [12,21], and from other parts of the Russian Far East [12,13,22,23] to the Baikal rift zone [24]. The genus' range also includes Sakhalin and the Kurile Islands [5,14,25–27]. In the southern part of Sakhalin Island, representatives of another amnicolid genus, *Akiyoshia* Kuroda and Habe, 1954, were found [12], while another species of this genus was described from a small pool in limestone caves in Japan [28].

Shell drawings of the type specimens of *Kolhymamnicola* species and identification keys have been repeatedly published [5,12,13]; however, the identification of its species remains complicated due to the unavailability of photographs of the type specimens.

To improve the morphological diagnosis of the genus, light shell photographs of the types and non-type specimens, male and female reproductive organs, and SEM images of protoconchs, operculums, and radular teeth were studied for all four known *Kolhymamnicola* species. We also conducted a molecular phylogenetic analysis to identify the nearest living relative of *Kolhymamnicola* and, on that basis, compare the morpho-anatomy of the closely related genera of the Amnicolinae. The new molecular data obtained in this study allowed us to propose some changes to the current system of the Amnicolidae s. lato. Another aim of this research was to study and visualize the distribution of *Kolhymamnicola* over its extensive range, including the previously unknown localities of these snails. The first estimates of

the age of evolutionary divergence of *Kolhymamnicola* and some other amnicolid taxa were obtained in this research. This allows us to discuss the fossil records to understand the origin of the genus *Kolhymamnicola*, and to reconsider the taxonomic status of some extinct genera previously assigned to the family Baicaliidae, and reject their close affinity to the living species of this group.

## 2. Material and Methods

The examined material mostly comes from the mollusk collection of the Federal Scientific Center of the East Asia Terrestrial Biodiversity, the Far Eastern branch of the Russian Academy of Sciences (Vladivostok, Russia, IBSS). We examined and photographed the type specimen (syntype) of *K. miyadii* kept in the National Science Museum, Tokyo (NSMT, Japan), and the holotypes and paratypes of the remaining three species (*K. kolhymensis*, *K. ochotica*, and *K. wasiliewae*) deposited in the continental mollusks collection of the Zoological Institute of the Russian Academy of Sciences (ZIN RAS, St. Petersburg, Russia).

Table 1 shows a list of the studied material, including samples collected by the authors during this work, kept in the Limnological Institute SB RAS, Irkutsk (LIN) and in the Laboratory of Macroecology and Biogeography of Invertebrates, St.-Petersburg State University (LMBI).

Prior to dissections, the shells were photographed using a Canon EOS 60D camera with a Canon MP-E 65 mm f/2.8 1–5$^\mathrm{x}$ macro lens. The same digital camera was used to take photographs of the type specimens of the fossil gastropod species housed in the collection of the Russian Geological Research Institute (VSEGEI, St. Petersburg, Russia). We examined specimens of 28 fossil species identified by G. Martinson as belonging to the families Hydrobiidae, Bithyniidae, and Baicaliidae sensu lato. These specimens were collected in Northern Asia, the Baikal rift zone, Mongolia, and China. Their fossil range varies from the Late Cretaceous to the Paleogene.

Protoconchs, radular teeth, operculums, and penises were examined using a scanning electron microscope (Quanta 200, USA). Prior to investigation, the objects were rinsed in chlorine bleach, washed with distilled water and alcohol, dried, mounted onto SEM stubs, and sputter-coated with gold. Penises, after being washed with 70 and 100% alcohol, were placed in hexamethyldisilazane for 10 h, then dried and mounted onto SEM stubs [29]. Shells were measured according to the standard scheme [5], and the measurements were performed on photographs using ImageProPlus for Windows.

In morpho-anatomical descriptions, we followed the terminology used in the works of Radoman [30], Szarowska [31], and Hershler [32–34].

We compared the morphology of *Kolhymamnicola* with that of some other genera of the Amnicolidae s. lato. as it is described in the literature: the European *Marstoniopsis* V.R. Altena, 1936 [=*Parabythinella* Radoman, 1973] [31], the Nearctic *Taylorconcha* [32,33] and *Amnicola* [34], as well as the East Asian taxa *Akiyoshia* [35]), *Erchaia* [35,36], and Baicaliidae, endemic to Lake Baikal [4,37–40]. Additionally, we used our own unpublished data on the morpho-anatomy of Baicaliidae species (19 specimens/species of 8 genera). Molecular analysis was performed using 8 juvenile individuals of *K. wasiliewae* from the Bolshoy In River, a tributary of the Tunguska River (Figure 1, locality 18).

DNA was extracted from the muscle tissue according to a protocol described by Sokolov [41].

**Table 1.** List of the studied material, including museum lots and new samples obtained during this research.

| Species | Locality (Numbers Correspond to Names in Figure 1 | Collection, Museum Acronym, Registration Number | Examined Specimens |
|---|---|---|---|
| *K. kolhymensis* | (1) Lower Kolhyma River basin, near Chersky settlement, 68°44′ N 161°21′ E (type locality) | ZIN, No. 1, 4 IBSS, No. 1418 | Holotype and 2 paratypes; 5 dry shells 2 alcohol-fixed females (juvenile and mature) |
| | Kolhyma River basin, lakes | ZIN, No. 6 | 2 dry shells (paratypes) |
| | (2) Middle Kolhyma River basin, an unnamed floodplain thermokarst lake, 66°39′ N 152°35′ E | IBSS, No. 2104 | 1 juvenile female |
| | (7) Chukotka Peninsula, a small lake, the Avtaatkui River basin, near Anadyr city, | IBSS, No. 1397 | 4 dry shells |
| | (12) Coast of the Okhotsk Sea, Perevolochnaya Bay, Yama River valley, an unnamed lake, 59°43′ N 149°46′ E. | IBSS, No. 2120b | 3 juvenile females, 1 male |
| *K. ochotica* | (14) Okhota River basin, Lake Bolshoe Uyeginskoe, 59°20′ N 143°01′ E (type locality) | ZIN, No.1 | Holotype (dry shell) |
| | (13) Kava River basin, coast of the Okhotsk Sea, 59°38′ N 149°05′ E | IBSS, No. 4325 | 1 juvenile female, 2 males |
| | (13) Yama River basin, Lake Tynerynda, 59°46′ N 149°23′ E; | IBSS, No. 2431 | 1 alcohol-fixed female, 3 dry shells |
| *K. wasiliewae* | (16) Lower Amur River basin, Bolin River (backwater), vicinity of Amursk Town, 50°19′ N 136°48′ E (type locality). | ZIN, No. 1; IBSS, leg. L. Prozorova, 8 July 1980. | Holotype, 1 juvenile female and 1 male, 3 dry shells |
| | (18) Bolshoy In River, a tributary of the Tunguska River, 48°36′ N 133°39′ E. | IBSS, leg. V. Makarenko & L. Prozorova | 8 juveniles (used in the molecular analysis) |
| | (20) Leprindo Lake, 56°37′ N 117°31′ E | LIN, leg. A. Matveev, 1997 | 1 male |
| | (22) Vicinity of Vitimsky Nature Reserve, lake in the Lapagar terrain, 57°15′ N 116°24′ E. | LMBI, No. 05-032 | 6 dry shells, 1 alcohol-fixed female |
| | (15) Upper Zeya River basin, 50°14′ N 127°34′ E. | IBSS, leg. L. Prozorova | 1 dry shell |
| *K. miyadii* | (11) Kurile Archipelago, Shumshu Island, Lake Bolshoye, 50°45′ N 156°15′ E (type locality). | ZIN, No. 1 and 2 | Holotype, 3 paratypes |
| | The same locality as above | IBSS, No. 111-97. | 4 dry shells, 3 alcohol-fixed specimens (2 females, 1 male) |
| | (23) Sakhalin Island, a floodplain lake near Tymovski settlement, the Tym River basin, 51°39′ N 142°56′ E; | LMBI, No. 05-031 | 4 fixed specimens |
| | (27) Kurile Archipelago, Zelenyi Island, Lake Sredneye, 43°29′ N 146°08′ E | IBSS, No. 518 | 5 dry shells, 5 fixed specimens (1 female, 4 males) |

Gene fragments of the mitochondrial cytochrome c oxidase subunit I (COI), mitochondrial large rRNA (16S), and nuclear small rRNA (18S) were amplified in PCR using the primers listed in Table S1. From 1 to 3 μL of purified DNA was amplified in a 25 μL reaction mixture using the BioMaster HS-*Taq* PCR Kit (Biolabmix, Russia) following the manufacturer's recommendation. We amplified all gene fragments using a temperature profile of 94 °C 4 min (94 °C 1 min, 50 °C (55 °C for 18S rRNA) 1 min, 72 °C 1 min) × 30 cycles, 72 °C 5 min. The amplicons were visualized on 1% agarose gel. Sequencing was performed using an ABI 3130 automated sequencer (Research and Production Company "SYNTOL", Moscow, Russia). The nucleotide sequences were verified manually and aligned using default settings in CLUSTAL W [42], implemented in BIOEDIT v.7.2.5 [43]. The resulting COI nucleotide alignment was translated to amine acids sequences to ensure that stop codons

were absent. The 16S and 18S nucleotide sequences were aligned with MAFFT v. 6.2 (e-ins-I algorithm) [44]. The obtained nucleotide sequences were compared using BLAST with those for species belonging to 6 genera of the Amnicolidae and 19 species of 8 genera of the Baicaliidae accessible from GenBank. Accession numbers of the new and retrieved nucleotide sequences are presented in Table S2. Maximum Likelihood (ML) phylogeny was inferred using IQ-TREE v.1.6.8 and midpoint rooted [45]. The most suitable model of molecular evolution was chosen using the Model Finder module within IQ-TREE [46]. Branch support was assessed using bootstrap values [47] and the Shimodaira–Hasegawa approximate likelihood ratio test (SHaLRT; see [48]). The time of divergence of *K. wasiliewae* from the most recent common ancestor (tMRCA) was estimated using the COI sequence dataset with a Bayesian approach with BEAST 1.10.4 [49]. The analysis was performed using a GTR+I+G model of nucleotide substitution selected using JMODELTEST. The suggested divergence rate was set to 1.96% per Myr, following Wilke et al. [50]. The Birth–Death model [51] was applied to a speciation process for our data set. The analysis was run for 100,000,000 generations, with the first 10,000,000 discarded as burn-in and parameter values sampled every 10,000 generations. The results were analyzed with TRACER 1.6 [52] to assess the convergence and confirm that the combined effective sample sizes for all parameters were larger than 200, ensuring that the Markov Chain Monte Carlo (MCMC) had run long enough.

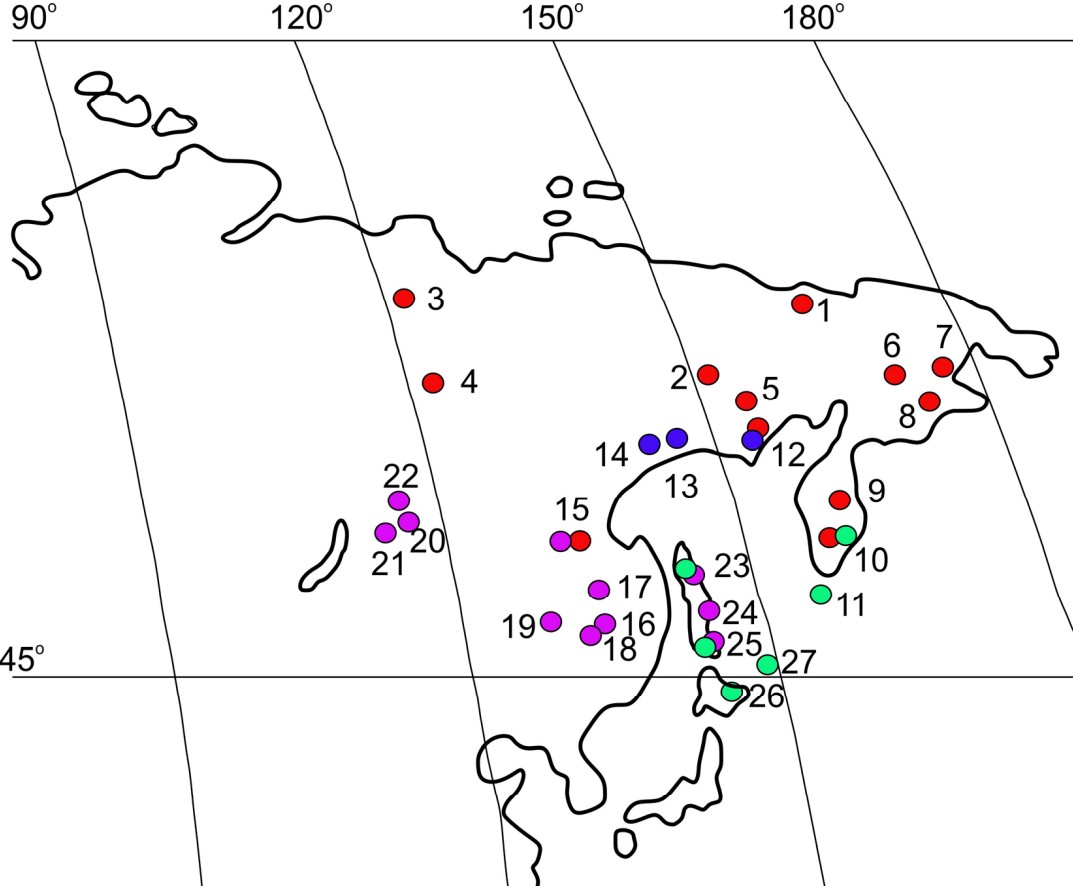

**Figure 1.** Distribution of *Kolhymamnicola* snails. Blue circles—*K. ochotica*, red—*K. kolhymensis*, violet—*K. wasiliewae*, green—*K. miyadii*. (**1**) Lower Kolhyma River basin, near Chersky settlement, 68°44′ N 161°21′ E (type locality). (**2**) Middle Kolhyma River basin, nameless floodplain-thermokarst lake, 66°39′ N 152°35′ E. (**3**) Lower Lena River basin, 67°49′ N 123°06′ E. (**4**) Lower Lena River basin, near the confluence of the Vilyui River, 64°27′ N 126°06′ E. (**5**) Middle Kolhyma River basin, Lake Priyatnoe, 62°07′ N 149°05′ E. (**6**) Chukotka, Anadyr River basin, nameless floodplain lakes (Nos. 5, 6) 64°59′ N 171°26′ E. (**7**) Chukotka, Avtotkui River basin, small floodplain lake, near Anadyr

city, 64°43′ N 177°31′ E. (**8**) Chukotka, Khatyrka River basin, Lake Rybnoe, 62°03′ N 175°16′ E. (**9**) Kamchatka, Lake Azabachye, 56°08′ N 161°47′ E. (**10**) Kamchatka, Lake Turpanka 52°56′ N 158°24′ E. (**11**) Kuril Islands, Shumshu Island, Lake Bolshoye, 50°45′ N 156°15′ E (type locality). (**12**) Coast of the Okhotsk sea, Perevolochnaya Bay, Yama River valley, nameless lake, 59°43′ N 149°46′ E. (**13**) Tynerynda lake 59°46′ N 149°23′ E; Kava River basin, coast of the Okhotsk sea, 59°38′ N 149°05′ E; Kava River basin, Sbornoe lake, 59°37′ N 147°07′ E. (**14**) Okhota River basin, lake Bolshoe Uyeginskoe, 59°20′ N 143°01′ E (type locality) (**15**) Upper Zeya River basin, 50°14′ N 127°34′ E. (**16**) Lower Amur River basin, Bolin River (backwater), vicinity of Amursk, 50°19′ N 136°48′ E (type locality). (**17**) Amgun River basin, Chukchagirskoye Lake, 51°57′ N 136°31′ E. (**18**) Bolshoy In River, tributary of Tunguska River, 48°36′ N 133°39′ E. (**19**) Middle Bureya River basin, 49°26′ N 129°30′ E. (**20**) Leprindo Lake, 56°37′ N 117°31′ E. (**21**) Tsypa Lake 54°50′ N 112°25′ E. (**22**) vicinity of the Vitimsky Nature Reserve, lake in the Lapagar terrain, 57°15′ N 116°24′ E. (**23**) Sakhalin Island (north part), lake connected to a tributary of Val River, 52°21′ N 143°04′ E; Tym River basin, flowing lake, and oxbow lake of the Tym River 51°39′ N 142°56′ E and near: floodplain lake near Tymovski settlement (**24,25**) channel between Ayskoye and Baklaniye lakes, 46°36′ N 143°18′ E; pond, connected with Lake Bolshoye Vavayskoye, 46°36′ N 143°18′ E. (**26**) Kunashir Island, Yuzhno-Kurilsk, Serebryanka River and Serebryanka lake, 44°02′ N 145°51′ E. (**27**) Zelyeniy Island, Lake Sredneye, 43°29′ N 146°08′ E.

## 3. Results

### 3.1. Morphology of Kolhymamnicola

The updated and extended morphological characteristics of the genus *Kolhymamnicola* are as follows.

Shell (Figure 2) small, up to 4 mm in height, light-brown, brown, corneous-brown, or green-brown, smooth with fine growth lines, ovoid-conical or conical depending on the whorl number. Whorls 3.5–5.0, ovate or rounded, in some individuals slightly shouldered, slowly increasing, with deep suture. Aperture ovate or rounded, parieto-palatal angle rounded or ovoid, obtuse angulated. Umbilicus wide or narrow slit [13]. Table 2 summarizes the interspecific differences in shell characteristics of the four nominal species of *Kolhymamnicola*. The holotype sizes, as well as the sizes of other individuals examined by us, are provided in Table 3.

Shell characters of the four species appeared to be rather variable, which makes accurate species identification almost impossible. We could not find specimens fully corresponding to the syntype of *K. miyadii* (Figure 2E) and the description published by Habe [14] among the topotypes of this species.

Operculum thin, paucispiral, nucleus submarginal, something smaller than aperture size, muscle scar oval-rounded slightly more than the nucleus (Figure 3A,B,D–F).

Protoconch slightly spiral sculptured or smooth, with a few thin spiral lines visible mainly near suture, whorl number from 1.25 to 1.5, diameter 520–780 μm, initial plate ~140–190 μm (n = 8) (Figure 3G–K).

Radular teeth row varies from 44 to 46 number, radular ribbon is ~390–450 μm length and 84–95 μm width. Central tooth is trapezoidal or broadly trapezoidal, its width 2.3 times exceeds the length, the basal tongue is V-shaped, and its length is almost equal to lateral margins (Figure 4). Central tooth formula is $\frac{(3-5)-1-(3-5)}{(1-2)-(1-3)}$. Lateral tooth (L) rectangular, its length exceeds its width, angled, 3–1–3, its central cusp is small, pointed, with 2–3 inner and 3 outer cusps, basal tongue poorly developed. Inner marginal (MI) teeth with broad cutting edge carrying about 18–20 cusps. Outer marginal teeth (MII) with 10–16 small cusps with rounded cutting edge. Habe [14] mentioned about 30-min cusps on the outer marginal. No interspecific differences in radular morphology can be observed (Table 2); the variation in the number of cusps within a single radular ribbon corresponds to that observed in individuals of the same species (Figure 4D,Da).

Soft body anatomy. The soft body is pale-yellow, with black pigment on the snout and along the edges of the tentacles (Figure 5Da,Db,E). Gill and osphradium without pigmentation, gill with broadly triangular filaments, 24–32 in numbers, osphradium wide,

slightly posterior or middle positioned, about 1/3 times shorter than gill (Figure 3C). Cerebral ganglions are dark-grey pigmented. Style sac length is less than stomach length. Digestive gland slightly brown with a single opening to the stomach. Caecum is absent.

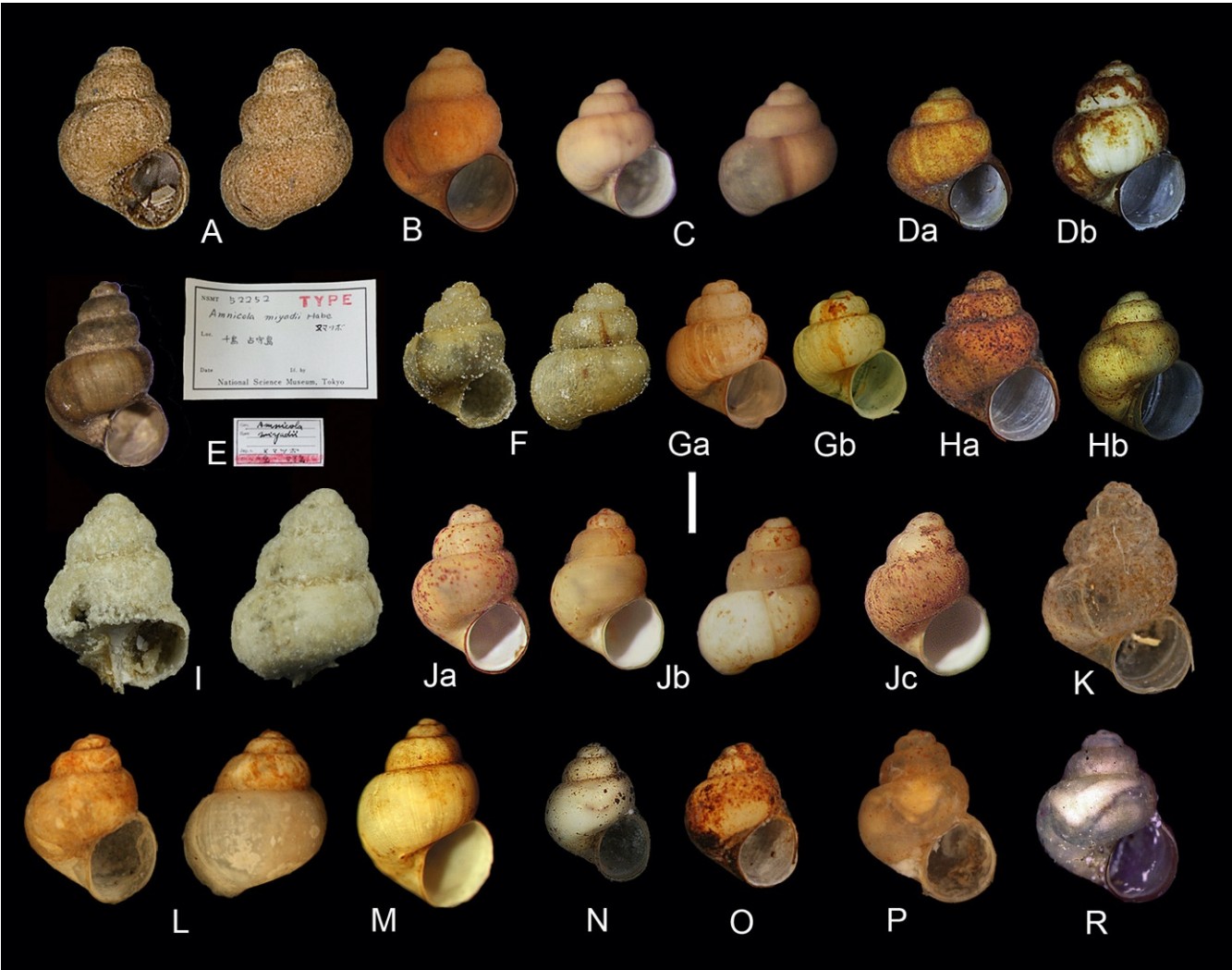

**Figure 2.** Shells of *Kolhymamnicola* species. (**A**) *K. kolhymensis*, paratype, ZIN No. 4. (**B**) *K. kolhymensis*, topotype, female, Lower Kolhyma River, near Chersky settlement, IBSS, No. 1418. (**C**) *K. kolhymensis*, female, an inlet of Lake Yama, Perevolochnaya valley, IBSS, No. 2120b. (**Da,Db**) *K. kolhymensis*, a lake near Anadyr city, Chukotka, IBSS, 1397. (**E**) *K. miyadii*, the type with the labels, NSMT. (**F**) *K. miyadii*, topotype, Bolshoe Lake, Shumshu Island, ZIN. (**Ga**) *K. miyadii*, topotype, female, the same lake, IBSS, No. 111-97. (**Gb**) *K. miyadii*, topotype, male, the same sample. (**Ha**) *K. miyadii*, female, Sredneye Lake on Zelenyi Island, IBSS, No. 518. (**Hb**) *K. miyadii*, male, the same sample. (**I**) *K. ochotica*, holotype, ZIN, No. 1. (**Ja**) *K. ochotica*, female, Kava River, Magadan Region, IBSS, No. 4325. (**Jb, Jc**) *K. ochotica*, males, the same locality. (**K**) *K. ochotica*, female, Lake Tynerynda, Yama River basin, IBSS, No. 2431. (**L**) *K. wasiliewae*, holotype, ZIN, No.1. (**M**) *K. wasiliewae*, female, topotype, Bolshoi Bolin River, near Amursk Town, IBSS, leg. L. Prozorova. (**N**) *K. wasiliewae*, male, Bolshoy In River, Amur River Basin, IBSS, leg. L. Prozorova. (**O**) *K. wasiliewae*, male, Upper Zeya River basin, IBSS, leg. L. Prozorova. (**P**) *K.* cf. *wasiliewae* Lake Leprindo, Baikal Rift Zone, male, LIN, leg. A. Matveev. (**R**) *K.* cf. *wasiliewae*, female, lake in the Lapagar terrain, vicinity of the Vitimsky Nature Reserve, LMBI, No. 05-031. Scale bars 1 mm.

Female reproductive system. Ovary consists of the simple lobes, albumen gland in dorsal view is longer than capsule gland. Coiled oviduct represents a simple colorless loop; its right part is sometimes expanded. Starobogatov and Budnikova [11] thought that the right part of the loop functions as a bursa; however, our observations do not support this opinion. Seminal receptacle poorly distinguished in some dissected females (Figure 5Ba), not pigmented, pearly-colored due to the sperm inside, the tubular opening at proximal part of the simple coil loop (rs$_2$ position, according to Radoman [30]), a tip of the seminal receptacle narrow or wide, straight, bent to the right or left (Figure 5C,G and Figure 6A–C). Ventral channel (or groove) goes along lateral part of the capsule gland (Figures 5G and 6A).

**Table 2.** Morphological and anatomical characteristics of *Kolhymamnicola* species, adapted from [13,14] and our own data.

| Species/Character | *K. kolhymensis* | *K. wasiliewae* | *K. ochotica* | *K. miyadi* | *K. miyadi* |
|---|---|---|---|---|---|
| | | | **[13]** | | **[14]** |
| Shell: | | | | | |
| shape | elongated conical | conical | ovoid-conical | ovoid | conical |
| height | 2.75–3.25 mm | 4.0 mm | 4.0 mm | <3.25 mm | 3.1 mm |
| color | green-brown | green or yellow-green | light brown | yellow, brown, or green | corneous brown |
| whorls number | 3.74–4.25 | 3.5–4.0 | 3.5–4.0 | 3.5–4.5 | 5.0 |
| whorl shape | rounded, slightly shouldered | ovate | rounded, shouldered | rounded, slightly shouldered | conspicuously inflated |
| spire height/shell height (Sph/SH) | ~0.6 | <0.5 | <0.5 | >0.5 | >0.5 |
| Aperture: shape | ovate | ovate | ovate | rounded | oblique, rounded pyriform |
| parieto-palatal corner | rounded | rounded, slightly narrowing | rounded, slightly narrowing | ovoid, obtuse angulated | slightly angled |
| Umbilicus | wide slit | narrow slit | slit, slightly covered by columellar lip | wide slit | very narrow perforated |
| | | | **Own Data** | | **[14]** |
| Species/Character | *K. kolhymensis* | *K. wasiliewae* | *K. ochotica* | *K. miyadi* | *K. miyadi* |
| Formulae of the central radula tooth | $\dfrac{(5)4-4(5)}{(2)1-1(2)}$ | $\dfrac{(4)3-1-3(4)}{(2)1-1(2)}$ | $\dfrac{(4)3-1-3(4)}{(2)1-1(2)}$ | $\dfrac{(4)3-1-3(4)}{(2)1-1(2)}$ | $\dfrac{(4)3-1-3(4)}{(3)2-2(3)}$ |
| Formulae for other teeth (see the text) | L (3–1–3) + MI (20) + MII (15) | L (3–1–3) + MI (~18-20) + MII (8–10) | L [3–1–2-(3)] + MI (~18) + MII (12–16) | L (3–1–3) + MI (20) + MII (10) | L (7) + M (~30) |
| Penis proximal base | slightly broad | narrow | broad | narrow | no data |
| Penis lobe and distal part of a penis | rather equal or ~2 times shorter | ~3 times shorter than distal part | ~2 times shorter than distal part | ~3–3.5 times shorter than distal part | no data |
| Penis tip | no papilla | with short papilla | no? papilla | no data | no data |
| Seminal receptacle | straight | turns right | turns left or nearly straight | turns left | |

**Table 3.** Shell size (in mm) of the *Kolhymamnicola* snails, syntype of *K. miyadii* adapted from [14], holotypes of 3 other species adapted from [13], and own data maked *, mean ± st. dev. (minimum–maximum), in mm.

| Species | SH (Shell Height) | SW (Shell Width) | SpH (Spire Height) | AL (Aperture Length) | AW (Aperture Width) | Whorl Number |
|---|---|---|---|---|---|---|
| *K. kolhymensis* | | | | | | |
| holotype | 2.75 | 2.2 | | 1.5 | 1.05 | 3.7 |
| (n = 6) * | 2.75 ± 0.27 (2.3–3.0) | 2.03 ± 0.14 (1.8–2.2) | 1.52 ± 0.23 (1.2–1.8) | 1.13 ± 0.14 (1.0–1.3) | 1.15 ± 0.10 (1.0–1.3) | 3.8 ± 0.3 (3.5–4.25) |
| *K. ochotica* | | | | | | |
| holotype | 4.2 | 2.6 | 2.4 | 1.8 | 1.4 | 4.0 |
| (n = 5) * | 2.78 ± 0.47 (2.2–3.5) | 2.09 ± 0.24 (1.9–2.5) | 1.59 ± 0.18 (1.0–1.3) | 1.30 ± 0.10 (1.2–1.4) | 1.14 ±0.11 (1.0–1.3) | 4.0 ± 0.3 (3.5–4.25) |
| *K. miyadii* | | | | | | |
| syntype | 3.1 | 2.2 | | 1.5 | 1.1 | 5.0 |
| (n = 12) * | 2.4 ± 0.32 (1.9–2.9) | 1.92 ± 0.20 (1.8–2.3) | 1.23 ± 0.24 (0.8–1.5) | 1.19 ± 0.12 (1.0–1.4) | 0.98 ± 0.10 (0.8–1.1) | 3.66 ± 0.31 (3.25–4.25) |
| *K. wasiliewae* | | | | | | |
| holotype | 2.7 | 2.05 | 1.3 | 1.4 | 0.9 | 4.0 |
| (n = 5) * | 2.56 ± 0.35 (2.2–3.0) | 1.94 ± 0.24 (1.6–2.2) | 1.38 ±0.18 (1.2–1.6) | 1.26 ± 0.21 (1.0–1.5) | 1.0 ± 0.14 (0.8–1.1) | 3.85 ± 0.33 (3.5–4.0) |

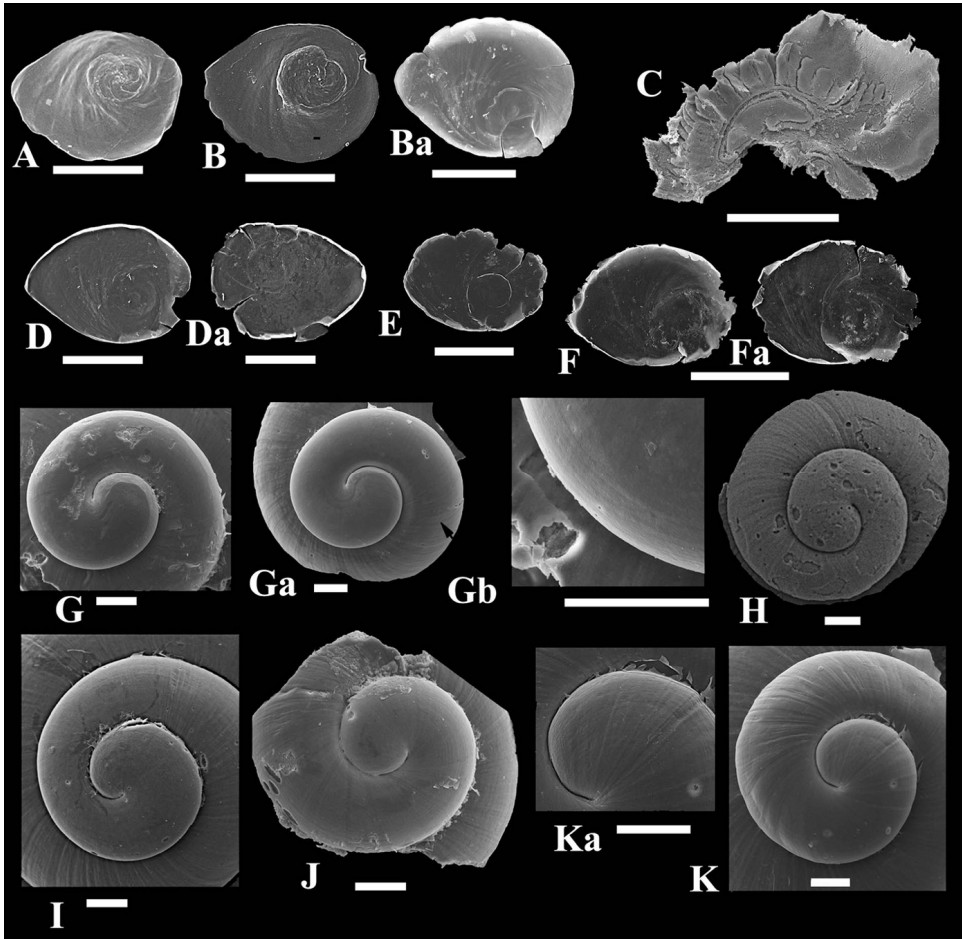

**Figure 3.** Operculums (**A**,**B**,**D**,**F**), gill with osphradium (**C**), and protoconchs (**G–K**) of *Kolhymamnicola* species. (**A**,**H**) *K. kolhymensis*, IBSS, No. 1418. (**B**, **C**,**G–Gb**) *K. ochotica* IBSS, No. 4325. (**F**,**Fa**,**I**) *K. wasiliewae*,

IBSS, leg. Makarenko & Prozorova. (**D,Da,K,Ka**) *K. miyadii* from Shumshu Island, ZIN, No. 2. (**E,J**) *K. miyadii* from Zelenyi Island, IBSS, No. 518. Scale bars: (**A–F**) 0.5 mm, (**G–K**) 100 μm.

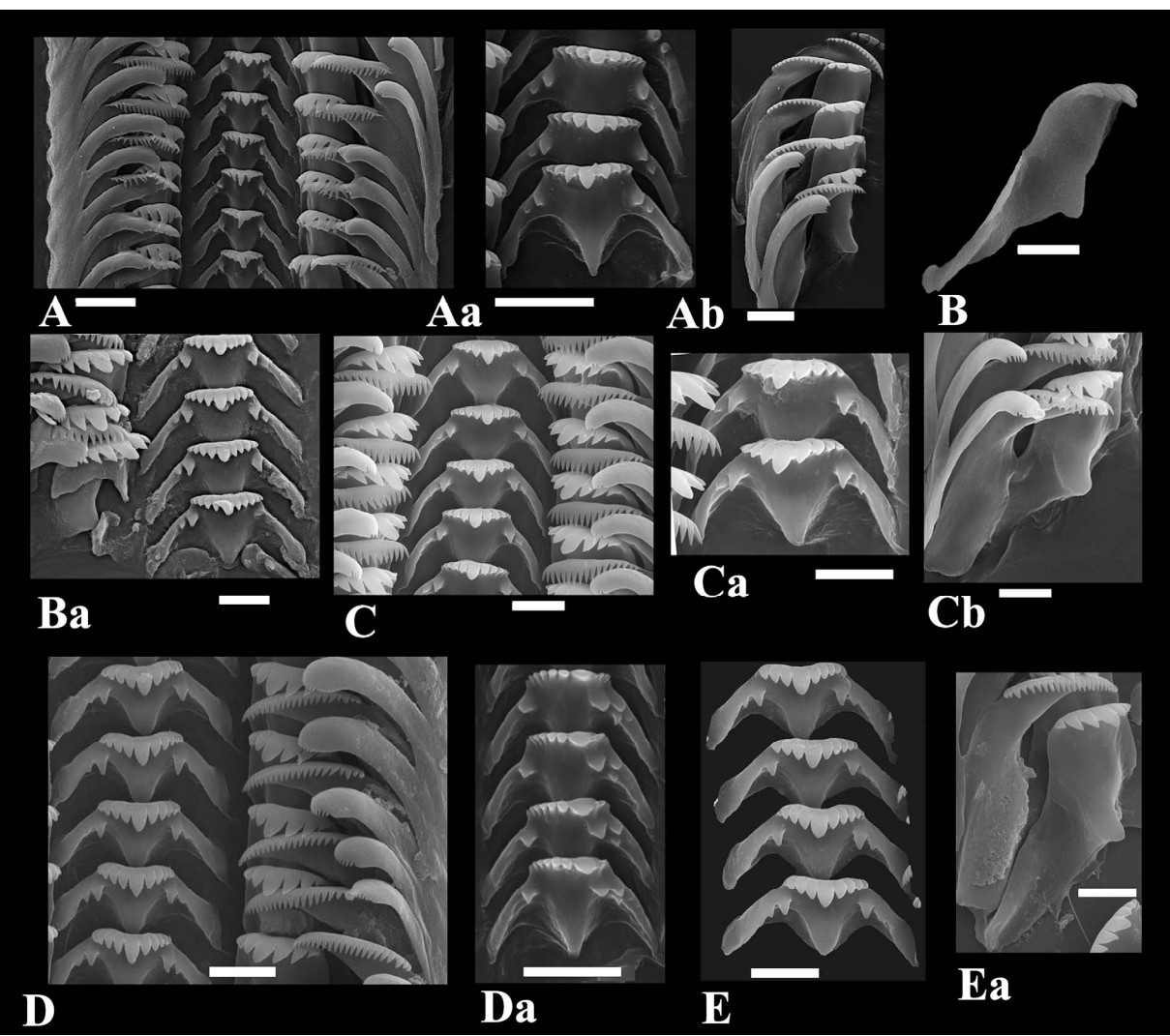

**Figure 4.** Radular teeth of *Kolhymamnicola* species. (**A,Aa,Ab**) *K. wasiliewae*, IBSS, leg. L. Prozorova, (**B,Ba**) *K. kolhymensis*, IBSS, No.2120b. (**C,Ca,Cb**) *K. ochotica*, IBSS, No. 4325. (**D,E**) *K. miyadii*, IBSS, No. 111-97. Scale bars 10 μm.

Male reproductive system. The testis branched, with long and large acini. Seminal vesicles massive, almost equal in size to the prostata. The prostata located in the posterior edge of the mantle cavity. The prostate duct emerges from the middle part of the gland; it is longer than prostate length. The base of the penis is gray-colored (*K. miyadii* from Shumshu Island) or without pigmentation (*K. miyadii* from Zelenyi Island, *K. wasilliewae* from Lake Leprindo) (Figure 5Da,Db,E,F). Penis bifid, attached basally to the right side of the midline part of the snout (Figure 5Db,E). Penial gland (or flagellum) outside penis in the form of a long tube rolled up into a ball (Figure 6D). The basal part of the penis equal in width to the middle part of the penis (*K. wasilliewae* and *K. miyadii*) (Figure 6D,E and Figure 7D,E) or dilated (*K. kolhymensis* and *K. ochotica*) (Figure 7A,B). Penial lobe length almost equal to distal part of the penis in *K. kolhymensis* or ~2 times shorter in *K. ochotica* and ~3.5 times shorter in *K. wasilliewae* and *K. miyadii* (Table 2). The opening of the penis rounded, simple,

or with a thickened edge and a short papilla (Figure 7Ca,Da). The penis morphology is virtually identical in *K. wasilliewae* and *K. miyadii*, without any clear differences.

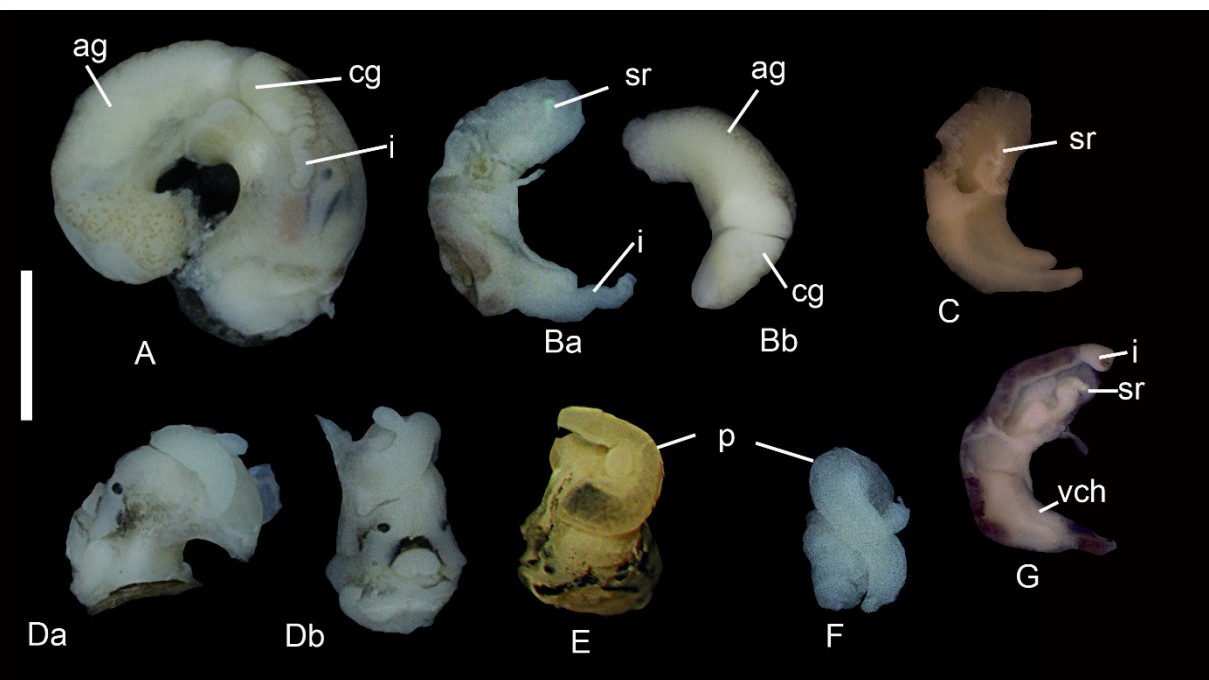

**Figure 5.** Soft bodies of *Kolhymamnicola*. (**A**) *K. miyadii*, female, the whole body, right side (shell, see Figure 1Ha). (**Ba**) *K. miyadii*, the same specimen, gonoduct, ventral view. (**Bb**) the same specimen, gonoduct, dorsal view. (**C**) *K. ochotica*, female gonoduct (shell, see Figure 1K); (**Da,Db**) *K. miyadii*, male (shell, see Figure 1Hb), the front part of the body with penis; (**Da**) the left side; (**Db**) frontal view; (**E**) *K. miyadii*, male, the front part of the body with penis (shell, see Figure 1Gb); (**F**) *K.* cf. *wasiliewae*, penis (shell, see Figure 1P); (**G**). *K.* cf. *wasiliewae*, female gonoduct (shell, see Figure 1R). Scale bars 1 mm. Abbreviations: ag—albumen gland, cg—capsule gland, i—intestine, p—penis, sr—seminal receptacle, vch—ventral channel.

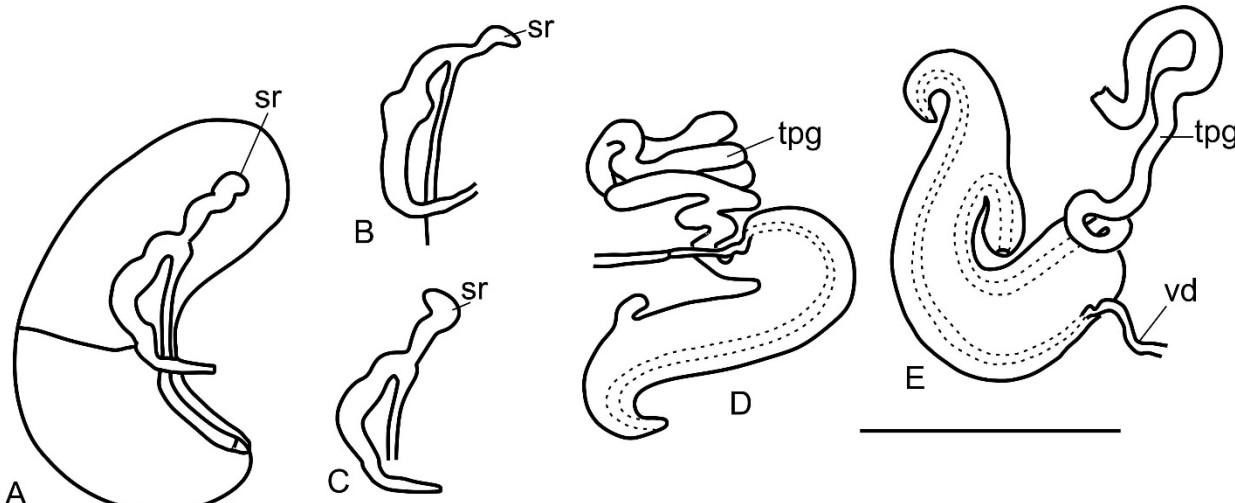

**Figure 6.** Drawings of reproductive organs of *Kolhymamnicola*. (**A**) Female gonoduct of *K. kolhymensis*. (**B**) a seminal receptacle of *K. ochotica.* (**C**) a seminal receptacle of *K miyadii* from Zelenyi isl., IBSS, No. 518 (**D,E**) penises of *K.* cf. *miyadii* (after Prozorova, original). Scale bars 1 mm. Abbreviations: sr—seminal receptacle, tpg—tubular penial gland, vd—vas deference.

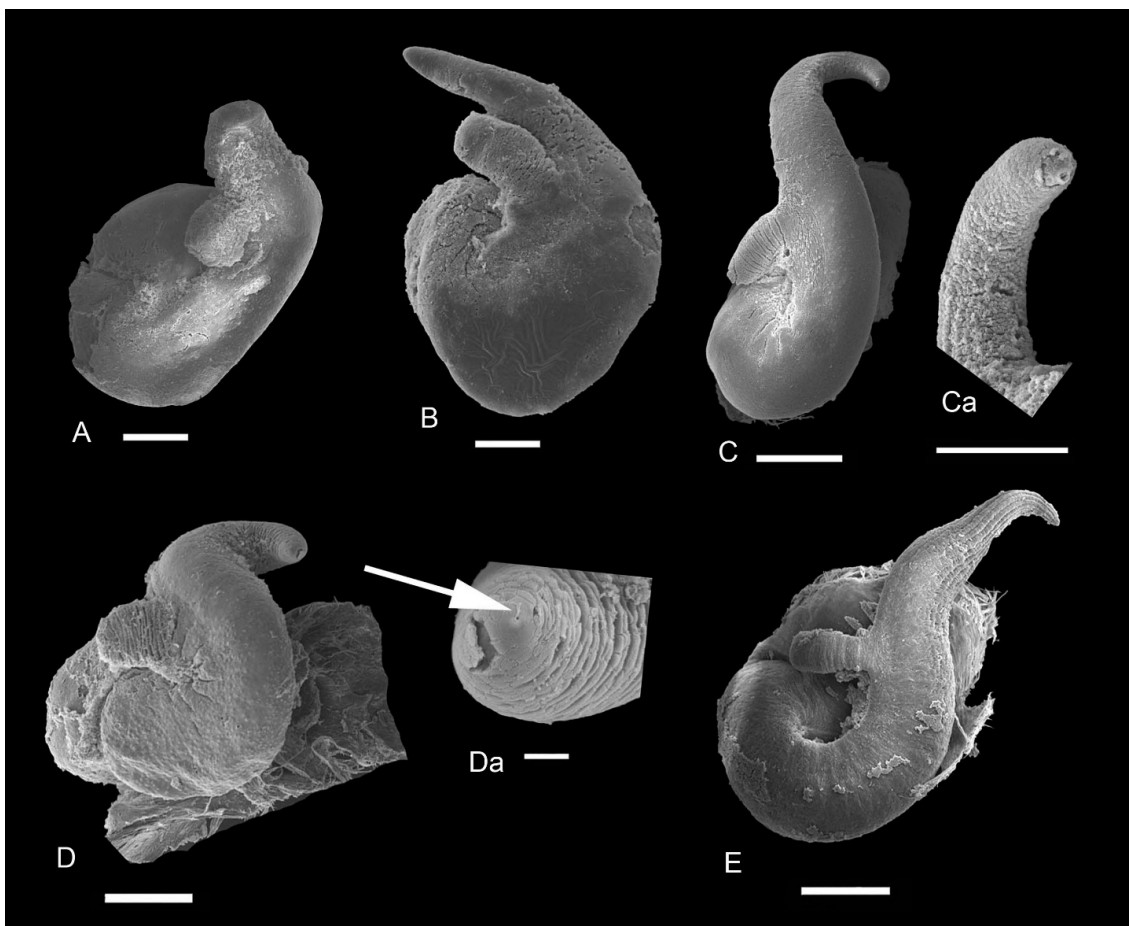

**Figure 7.** SEM photographs of penises of *Kolhymamnicola*. (**A**) *K. kolhymensis* from Perevolochnaya Valley, IBSS, No. 2120b. (**B**) *K. ochotica* from Kava River basin, IBSS, No. 4325. (**C**) *K. wasiliewae* from Bol'shoi Bolin River basin, IBSS, leg. L. Prozorova. (**Ca**) increasing tip of the penis; (**D**) *K. miyadii* from Zelenyi Island, IBSS, No. 518. (**Da**) increasing tip of the penis; (**E**) *K. miyadii* from Shumshu Island (shell, see Figure 1Gb). Scale bars: (**A**,**B**,**D**,**E**) 100 μm; (**Ca**) 50 μm, (**Da**) 10 μm.

*3.2. A Morphological Comparison between Kolhymamnicola and Other Representatives of the Amnicolidae s. Lato*

The female gonoduct in *Taylorconcha* is characterized by the absence of the bursa copulartix, the seminal receptacle has $rs_2$ position, and departs from the proximal part of the simple, small coiled loop; the ventral channel goes along the lateral part of the capsule gland [32,33]. The main difference between *Kolhymamnicola* and *Taylorconcha* lies in the penis morphology: the latter genus, unlike the former, has a simple penis (Table 4). According to Szarowska [31], *Marstoniopsis,* like *Kolhymamnicola,* possess a similar bifid penis with coiled glandular flagellum (penial gland); the female gonoduct of *Marstoniopsis* lacks a well-formed bursa, its long seminal receptacle ($rs_1$) has another position than in *Kolhymamnicola*. Moreover, females of *Marstoniopsis* have a ventral spermathecal duct (not a channel). Other morpho-anatomical differences between *Kolhymamnicola* and the genera *Erhaia*, *Akioshia*, and *Amnicola* are given in Table 4. The Baicaliidae are characterized by significant variability of the protoconch, its shape (from flat, valvatoid to turricate) and microsculpture (from smooth to strongly sculptured), the teleoconch shape, size, and sculpture, the numerous gill leaflets (up to 130), the radular teeth morphology (the central tooth without basal cusps and numerous cusps are present on the cutting edge of all teeth), absence of the well-formed bursa and seminal receptacle, function of which carry out a few or numerous diverticula of the coiled renal loop, the penis with an oval glandular part covering its basal side, and with short papilla on the tip.

**Table 4.** Comparative morphological characteristics of the genera *Kolhymamnicola*, *Taylorconcha*, *Erchaia*, *Marstoniopsis*, *Amnicola* and the family Baicaliidae.

| Characters | Taxa, Sources of Data | | | | | | |
|---|---|---|---|---|---|---|---|
| | *Kolhymamnicola* | *Taylorconcha* | *Marstoniopsis/Parabythinella* | *Erhaia* | *Akiyoshia/Saganoa* | *Amnicola* | **Baicaliidae** |
| | **[12,13], Own Data** | **[32,33]** | **[31]** | **[35,36]** | **[35,53]** | **[34]** | **[4,37,40], Own Data** |
| Shell height | up to 4 mm | up to 4 mm | <6 mm | $\leq$3 mm | <2 mm | up to 7 mm | 2–25 mm |
| Shell shape | ovate-conical or conical | globose to ovate-conical | ovate-conical | ovate-turreted or turreted | turreted | ovate-conical | ovate-conical, globose, conical, turreted, elongated-conical etc. |
| Shell whorl number | 3.5–5 | 3.5–4.5 | 4–4.5 | <5.5 | 4.5–5.5 | 4–6 | 3.5–12.0 |
| Shell sculpture | smooth | smooth | smooth | smooth | smooth | Smooth | smooth or sculptured (with ribs, carinae, spiral threads etc.) |
| Protoconch sculpture | smooth or with a few short spiral lines | fine spiral lines | delicate spiral lines | smooth with a few spiral lines near suture or spiral microsculpture | ?/smooth | smooth, sometimes with a few fine spiral lines | with spiral lines, striae or lirae |
| Operculum, position of the nucleus | submarginal | submarginal | submarginal | significant submarginal | significant submarginal | submarginal | submarginal or close to central position |
| Size and position of osphradium along ctenidium | 1/3 of gill length, posterior to middle position | slightly anterior to middle | ? | slightly posterior | ?/middle | $\frac{1}{4}$ of gill length, posterior | from 1/3 to 1/6 of gill length, anterior to middle |
| Number of ctenidium leaflets (filaments) | ~24–32 (n = 8) | ~17 | ? | <17 | ?/11–14 | 25–35 | 20–130 (n = 19) |
| Bursa copulatrix | absent | absent | absent | present | ?/present | Present | absent |
| Seminal receptacle | present, $rs_2$ | present, $rs_2$ | present, $rs_1$ | present, $rs_1$ | ?/absent, renal loop functions as seminal receptacle | present, $rs_1$ | absent, folded left part of the coil loop functions as seminal receptacles and bursa, some species with temporary $rs_1$ or/and $rs_2$ |

**Table 4.** *Cont.*

| Characters | Taxa, Sources of Data | | | | | | |
|---|---|---|---|---|---|---|---|
| | *Kolhymamnicola* | *Taylorconcha* | *Marstoniopsis/Parabythinella* | *Erhaia* | *Akiyoshia/Saganoa* | *Amnicola* | **Baicaliidae** |
| | [12,13], Own Data | [32,33] | [31] | [35,36] | [35,53] | [34] | [4,37,40], Own Data |
| Ventral channel (groove) or spermathecal duct | ventral channel | ventral channel | spermathecal duct | spermathecal duct | ?/spermathecal duct | spermathecal duct | ventral channel |
| Penis simple or with a lobe (bifid) | bifid | simple | bifid | simple | ?/simple | Bifid | simple |
| The accessory penial gland | present | absent | absent | absent | ?/absent | Present | penial gland is represented by a lamina on the penis base |
| Penis tip with or without papilla | with short or without papilla | without papilla | ? | with or without papilla | ?/with short papilla | ? | with or without papilla |
| Central radular tooth: shape | trapezoidal or broadly trapezoidal | broadly trapezoidal | broadly trapezoidal | trapezoidal | trapezoidal | broadly trapezoidal | square or rectangular |
| Shape of basal tongue | V-shaped | V-shaped | V-shaped | V-shaped | V-shaped | V-shaped | hollow, straight or slightly convex |
| Formulae of the central radular tooth | $\frac{(5)3-1-3(5)}{(2)1-1(2)}$ | $\frac{(5)4-1-4(5)}{1-1}$ | $\frac{3-1-3(4)}{(2)1-1(2)}$ | $\frac{(6-7)5-1-5(7)}{1-1}$ | $\frac{(5)4-1-4(5)}{(2)1-1(2)}$ | $\frac{(4)3-1-3(4))}{(3)2-2}$ | $\frac{(18)10-0(1)-10(18)}{0}$ |
| Distribution, typical habitats | Northern Asian mainland, Kurile islands, small lakes | Northwestern USA, rivers | Europe, Balkan peninsula, lakes | China, India, small streams | Japan, China, caves, drilled pipe wells, streams | Eastern part of the USA, creeks, swamps, lakes | Lake Baikal, various substrates and depths (from 1.5 to ~300 m) |

### 3.3. Molecular Phylogeny

In total, we obtained 8 fragments of 665 bp for the CO1 gene of *K. wasiliewae*, representing two unique haplotypes; 8 sequences of 510 bp for the mitochondrial 16S rRNA gene, belonging to a single haplotype and 4 nuclear 18S rRNA fragments of 900 bp long, also referring to a unique haplotype.

The mean pairwise *p*-distances based on COI sequences are given in Table 5. The genetic distance of 0.113 was found to separate *Kolhymamnicola* and *Taylorconcha* (*T. serpenticola* and *T. insperata*). These two closely related genera formed a single clade on the ML phylogenetic tree (Figure 8).

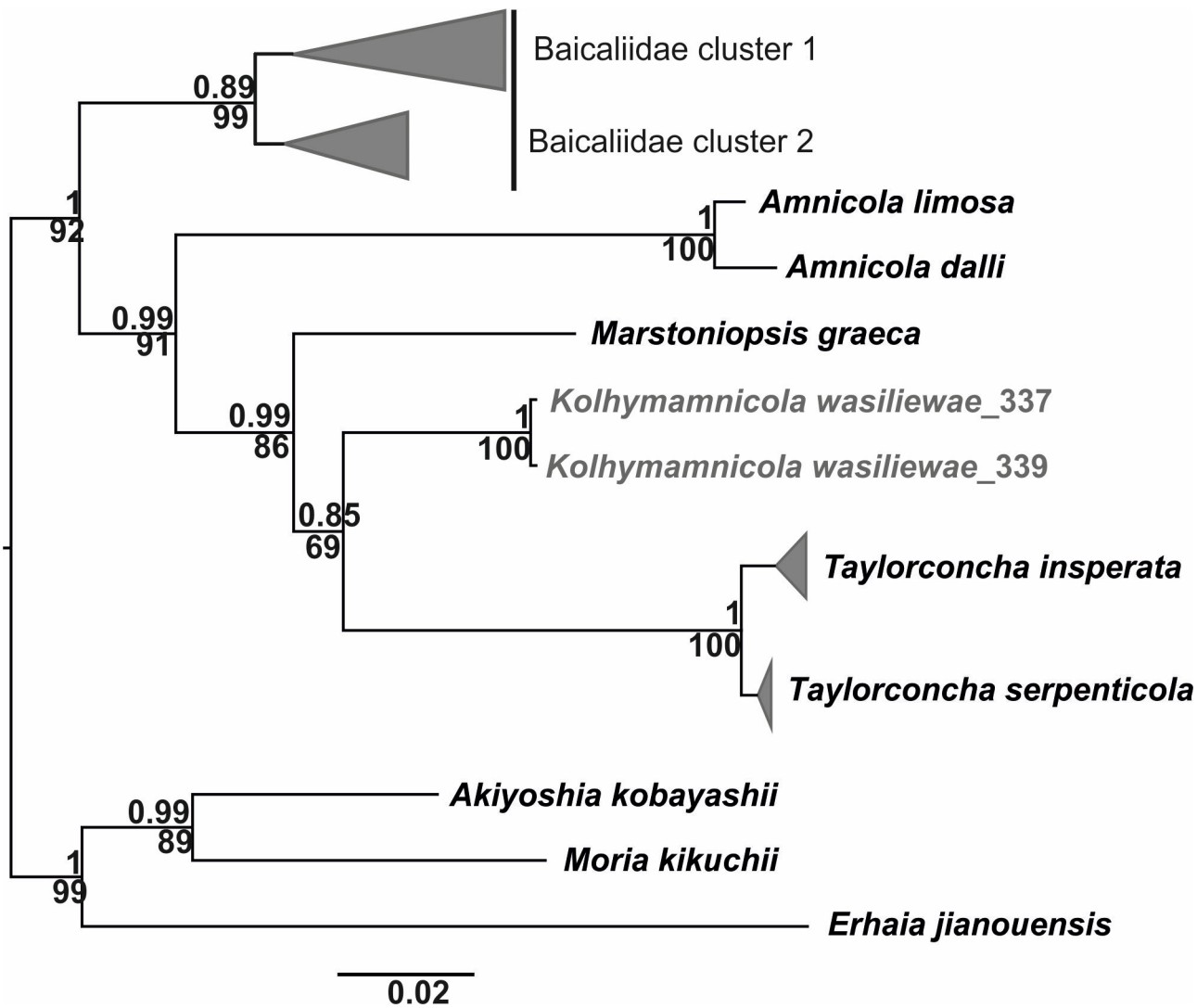

**Figure 8.** ML phylogenetic tree based on nucleotide sequences of COI, 16S, and 18S gene fragments. SH-aLRT supports are given above branches; bootstrap supports are given below branches.

The obtained phylogenetic tree based on combined nucleotide sequences for COI, 16S, and 18S gene fragments consists of two highly supported clades, one of them includes *Kolhymamnicola*, the North American genera *Taylorconcha* and *Amnicola*, as well the European *Marstoniopsis*. The Baicaliidae are paraphyletic in relation to this clade. Another clade comprises the Far East genus *Akiyoshia* and the Southern Asian genera *Moria* and *Erhaia*, the genetic distance between them was 0.099.

Results of the molecular phylogenetic analysis allow us to rank the three main phylogenetic lineages as the families: Amnicolidae, Baicaliidae, and Erhaiidae Davis and Kuo,

1985 **stat. nov**. The latter was originally established as a tribe within the family Pomatiopsidae [53] and then within Amnicolinae [2]. Our solution is furthermore supported by morpho-anatomical data (Table 4).

The estimated ages of divergence events within the Amnicolidae sensu lato, calculated according to COI-tree (not shown), revealed that the genera *Taylorconcha* and *Kolhymamnicola* are the closest relatives, their divergence took place about 5.5 Myr ago (95% CI: 3.3–7.8 Myr). The baicaliids separated from the clade, which includes European and North American genera, during the Miocene–Pliocene, at about 10.4 Myr (95% CI: 6.5–15.7 Myr). The divergence between Amnicolidae and Erchaiidae is dated 14.7 Myr (95% CI: 10.9–20.6 Myr).

**Table 5.** Pairwise uncorrected *p*-distances of COI sequences between different species of Amnicolidae and Baicaliidae.

| | Species | 1 | 2 | 3 | 4 | 5 | 6 | 7 | 8 | 9 | 10 |
|---|---|---|---|---|---|---|---|---|---|---|---|
| 1 | *Taylorconcha serpenticola* | | | | | | | | | | |
| 2 | *Taylorconcha insperata* | 0.012 | | | | | | | | | |
| 3 | *Kolhymamnicola wasiliewae* | 0.113 | 0.113 | | | | | | | | |
| 4 | *Marstoniopsis graeca* | 0.118 | 0.118 | 0.126 | | | | | | | |
| 5 | *Akiyoshia kobayashii* | 0.148 | 0.148 | 0.155 | 0.166 | | | | | | |
| 6 | *Erhaia jianouensis* | 0.159 | 0.159 | 0.162 | 0.154 | 0.142 | | | | | |
| 7 | *Moria kikuchii* | 0.155 | 0.152 | 0.160 | 0.159 | 0.099 | 0.145 | | | | |
| 8 | Baicaliidae cl. 1 | 0.142 | 0.144 | 0.145 | 0.137 | 0.161 | 0.138 | 0.162 | | | |
| 9 | Baicaliidae cl. 2 | 0.141 | 0.144 | 0.152 | 0.154 | 0.149 | 0.162 | 0.154 | 0.065 | | |
| 10 | *Amnicola limosa* | 0.164 | 0.169 | 0.172 | 0.184 | 0.174 | 0.162 | 0.171 | 0.159 | 0.156 | |
| 11 | *Amnicola dalli* | 0.160 | 0.166 | 0.172 | 0.186 | 0.167 | 0.171 | 0.174 | 0.154 | 0.154 | 0.027 |

*3.4. Kolhymamnicola Distribution and Habitats*

The distribution map of *Kolhymamnicola* (Figure 1) was compiled on the basis of an extensive set of literature data [10–14,16,20,21,24–26] and new findings made during this research. The range of this genus covers a large part of the North Asian mainland as well as the Chukotka and Kamchatka peninsulas, basins of the Kolhyma, Lena, Zeya, and Amur rivers, the Sea of Okhotsk coast, the northern and southern Kurile Islands, Sakhalin Island, lakes of the Baikal rift zone. The northern part of the range is inhabited mainly by *K. kolhymensis,* and the western boundary of its distribution runs along the Lower Lena River (Figure 1, locations 3, 4). *K. wasiliewae* occupies the southern part of the genus range. Localities of this species are known from the Amur River basin (Figure 1, locations 15-19), in lakes Leprindo and Tsypa, and in the Lapagar terrain of the Vitim River basin (Baikal rift zone) (Figure 1, locations 20, 21, 22). Snails found in lakes of Sakhalin Island (Figure 1, locations 23–25) were identified *K. wasiliewae* or *K. miyadi*. Lakes of the Kurile islands are inhabited by *K. miyadi* (Figure 1, locations 10, 11, 25–27). *K. ochotica* was found in lakes of the northern part of the Khabarovsk region (Tynerynda and Sbornoe lakes) as well as in waterbodies of the Yama River basin in the Magadan Region (Figure 1, localities 12–14).

Snails belonging to either *K. kolhymensis* or *K. wasiliewae* were recorded in a water body of the Upper Zeya River basin (Figure 1, location 15), *K. kolhymensis* and *K. miyadii* were found in the southern part of the Kamchatka Peninsula (Figure 1, location 10). Note that the snails from some locations cannot be identified exactly due to the overlap of the diagnostic features proposed for their identification. *Kolhymamnicola* snails mainly inhabit lakes of different origin, including the floodplain and thermokarst ones. We collected these gastropods in an unnamed lake of Lapagar terrain (vicinity of the Vitimsky Nature Reserve, Baikal rift zone) from the surface of aquatic vegetation (*Carex*, *Comarum*, *Calla*, *Menyanthes*) and soft substrates at the depth range 0–0.6 m. In this lake, as well as in its marshy parts, several species of pea-clams (*Euglesa*) were identified together with seven gastropod species: *Valvata sibirica* Middendorff, 1851, *Ampullaceana lagotis* (Schrank, 1803), *Peregriana dolgini* (Gundrizer and Starobogatov, 1979), *Gyraulus acronicus* (Férussac, 1807),

*Gyraulus stroemi* (Westerlund, 1881), *Kolhymorbis shadini* Starobogatov and Streletzkaja, 1967, and *Segmentina* cf. *nitida* (Müller, 1774).

Similar observations were made in August 2021 in a floodplain lake near the Tymovski settlement (Sakhalin Island). The accompanying malacofauna included *Boreoelona contortrix* (Lindholm, 1909), *Cipangopaludina kurilensis* Starobogatov, 1979, *Radix auricularia* (Linnaeus, 1758), *Gyraulus centrifugops* (Prozorova and Starobogatov, 1997), and *Euglesa* spp.

We did not reveal significant differences in the habitat ecology among four species of *Kolhymamnicola*.

## 4. Discussion

### 4.1. Species Composition and Radiation of the Genus Kolhymamnicola

The species content of *Kohlymamnicola* did not change after our research. After all, the available materials were not enough to secure the complete integrative revision of this genus. Most topotypic specimens used during this work were represented by juvenile or subadult individuals; this made it very difficult to use data on the anatomy of the reproductive system, which is traditionally considered an important source of taxonomic signal in many caenogastropod families. The molecular data are still accessible for a single species only (*K. wasiliewae*), and the number of specimens used in our morphological analyses was too low to undertake a statistical revision of the species boundaries. Due to the lack of representative samples of *K. wasiliewae*, we were unable to provide evidence of the sexual dimorphism in shell traits in this species. According to Bogatov and Zatrawkin [12,13], the males of *K. wasiliewae* have more elongated shells than the females; this difference is found among adult individuals with 4.25 and more shell whorls.

We found that the existing identification keys [5,13] sometimes are not helpful for reliable species identification, and, in certain studied samples, some individuals could not be assigned to either species unequivocally. Despite this, we refrain here from the synonymization of species with overlapping shell characters. The overlap of the morpho-anatomical characters (sometimes accompanied by distribution overlap) and the lack of substantial niche diversification revealed within *Kolhymamnicola* are a fairly common phenomenon in many groups of freshwater caenogastropod snails originated as a result of non-adaptive radiation. A good example is provided by the genus *Bythinella* Moquin–Tandon, 1856, endemic to the fauna of Europe [54]. Many species of *Bythinella*, whose species status is strongly confirmed by molecular data, cannot be identified based on their morphological traits [54–60]. The ranges of such cryptic species may often overlap. According to Wilke et al. [54], the diversification within *Bythinella* was "not accompanied by adaptation into various niches and resulting in a group of allopatric taxa." Falniowski and Szarowska [59], having conducted a complex morpho-anatomical and molecular genetic analyses of the Greece *Bythinella*, revealed that the radiation of the 10 species occurred through range fragmentation, with events of long-distance colonization and restricted gene flow with isolation by distance.

By analogy with the genus *Bythinella*, we can expect the presence of cryptic species in the genus under consideration (for example, in the Vitim River basin). On the other hand, it is not impossible that future research will show the identity between the two widely distributed mainland species with overlapping ranges (i.e., *K. kolhymensis* and *K. ochotica*).

In agreement with Habe [14], we identified the individuals of *Kolhymamnicola* collected in the northern and southern Kurile Islands as belonging to a single species, *K. miyadii*; however, some differences in certain shell and anatomical characteristics between snails of the Shumshu and Zelenyi islands are observed (Figure 2F,H and 5D,E). The Kurile archipelago has a long history dating back to the Late Oligocene and Miocene, but the island territories became isolated much later (7000–15,000 years ago) [61]. Thus, the problem of species composition of *Kolhymamnicola* on the Kurile Islands has not been definitely resolved.

*4.2. The Fossil Amnicolidae s. lato and the Origin of the Kolhymamnicola*

The results of this study show that the nearest living relative of *Kolhymamnicola* is the genus *Taylorconcha*, restricted in its distribution to the northwestern United States (Idaho). As the divergence of the two genera from their common ancestor took place in Pliocene, we can hypothesize that this ancestor belonged to a certain group of amnicolid snails, which in that geological epoch had a broad Holarctic range. The Bering land bridge, which had a major impact on both continents' ecosystems throughout the Cenozoic and was closed in the Pliocene, could secure biotic exchange between Eurasia and North America [62–65]. Probably, the fragmentation of this original range with the separation of its North Asian and North American parts promoted the further diversification of these two genera.

Another group that evolved from this ancestor was *Marstoniopsis*, a genus of European distribution. It is interesting that the bifid penis is the common character for both *Kolhymamnicola* and *Marstoniopsis*, while the structure of the female reproductive organs of *Kolhymamnicola* is similar to that of *Taylorconcha*. The COI-based genetic distances between these genera are almost equal: 0.126 between the two firsts, 0.113 between the two seconds, and 0.118 between *Taylorconcha* and *Marstoniopsis* (see Table 5). Possibly, the closeness of these distances means that during the divergence of these groups, asymmetric reproductive isolation (or asymmetrical secondary contacts) took place. The molecular traces of secondary contacts can be detected by estimating the effective population size and comparing the topologies of phylogenetic trees using several mitochondrial and nuclear molecular markers.

The identification of the probable common ancestor of these three genera among the known fossil representatives of Amnicolidae is difficult due to the scarcity of the paleontological record of this group in Northern Asia. So far, no fossil shells identified as Amnicolidae have been found either in the Baikal rift zone or in Northern Asia as a whole, including Mongolia [66–68]. However, Martinson [66] mentions that the shell of *Bithynia jurassica*, described by him from Jurassic deposits of Eastern Siberia, is similar to shells of the amnicolids described from the Morrison Formation (Jurassic of North America) by Yen [69]. We give a photo of an imprint of a shell of this species, as well as that of the entire shell of *B. leachioides* Martinson, 1961, found in Lower Cretaceous sediments of East Zabaikalie, including Vitim Plateau and Mongolia (Figure 9A,Ba,Bb). In our opinion, these species superficially resemble shells of *Kolhymamnicola* but differ from the latter by their larger sizes. It is impossible, however, to insist that these species of "*Bythinia*" are closely affined to the living Amnicolidae, let alone the genus *Kolhymamnicola*. Their external resemblance may well be a result of convergent evolution.

Though the fossil amnicolid snails in North America are known from such ancient deposits as Jurassic (Morrison Formation), Early (Bear River Formation), and Late Cretaceous (Wyoming, Utah, Colorado) ones [69–72], their affinity to the recent amnicolids looks very problematic, especially since the molecular estimates obtained in this research reveal the Miocene origin of the Amnicolidae. According to these estimates, no Mesozoic *Amnicola*-like fossil species can be placed in this family that originated much later. The same is true for numerous fossil "baicaliids" of Mesozoic and early Tertiary age (see below).

*4.3. Notes on the Erhaiidae stat. nov.*

The diagnosis of Erhaiidae stat. nov. corresponds to that of the tribe Erchaiini [53]. The family comprises genera *Akiyoshia, Moria,* and *Erhaia*; the morpho-anatomical differences between them are summarized in Table 4. *Erhaia* is a group, including today, up to 23 species [73] widely distributed in East and South Asia. In addition to these genera, a North Indian genus *Chencuia* Davis, 1997 [36] probably has to be placed here.

The genetic closeness of *E. jianouensis* to the genus *Akiyoshia* [73] is also confirmed by our results. The anatomical difference between these two genera lies in the absence of a separate seminal receptacle in *Akiyoshia*. Note that the genus *Akiyoshia* is represented by two subgenera—the nominative one and *Saganoa* Kuroda, Habe, and Tamu, 1957, whose

nucleotide sequences are absent from the GenBank. The origin of Erhaiidae **stat. nov.** dates back to Middle Miocene (see Section 3).

The range of Erhaiidae **stat. nov.** covers vast territories of South and East Asia, northwards to Sakhalin Island.

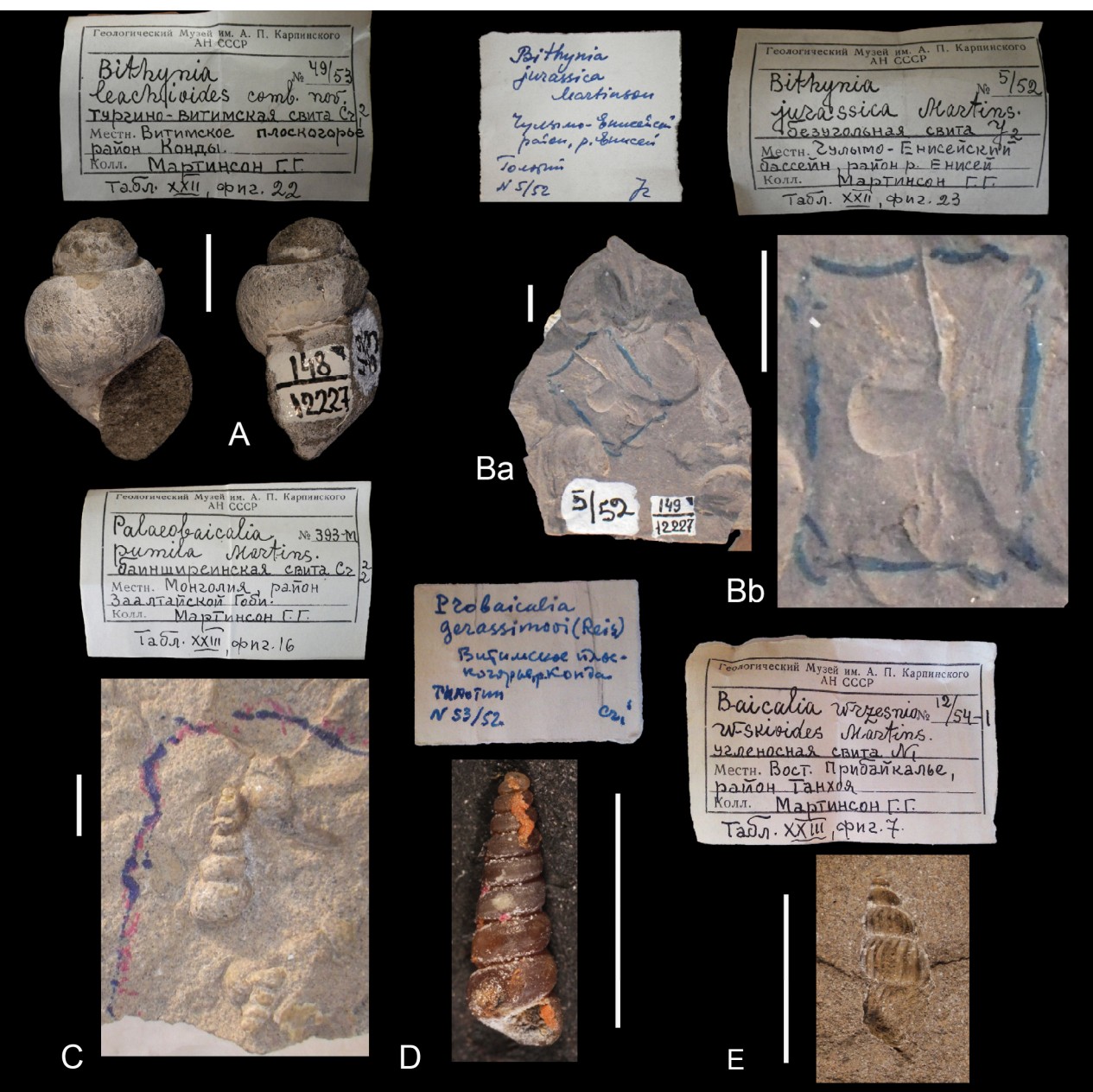

**Figure 9.** Some fossil representatives of 'Bithyniidae', Palaeobaicaliidae, and Baicaliidae with their museum labels. (**A**) *Bythinia leachioides* Martinson, 1961, holotype. (**Ba,Bb**) *Bythinia jurassica* Martinson, 1961, holotype. (**Ba**) Deposit with shell imprint. (**Bb**) the same, enlarged. (**C**) *Palaeobaicalia pumila* Martinson, 1961, holotype, the type species of the genus. (**D**) *Probaicalia gerassimovi* (Reis, 1910), topotype. (**E**) '*Baicalia*' *wresniowskioides* (Martinson, 1951). Scale bars 5 mm.

### 4.4. The Taxonomic Status of Recent and Some Fossil Representatives of Baicaliidae

According to our molecular phylogenetic analysis, the recent baicaliid snails constitute a highly-supported clade, paraphyletic in relation to the clade containing the genera *Amnicola*, *Taylorconcha*, and *Kolhymamnicola* (see Figure 8). Recent baicaliids display ecological niche diversification in a geographically isolated ancient lake and morpho-anatomical

uniqueness (see Table 4). They form a compact species flock, distributed exclusively in Lake Baikal and the lower courses of the Angara River [74–78]. One can consider this group as an outcome of a relatively recent radiation, whose age is estimated as roughly equal to ~3.5 Myr [76]. These facts, in our opinion, substantiate a family rank for Baicaliidae, as it is treated in Russian literature [4–7]. This family includes 8 recent genera [40,75] with several decades of biological species, the specific status of more than half of which is genetically confirmed, and several new species waiting for their description [75–81].

The taxonomic position of the genus *Pyrgobaicalia* Starobogatov, 1972 needs a re-evaluation. This genus was described from Tajikistan, Central Asia [82], and its placement into Baicaliidae or Amnicolidae is highly doubtful [83]. The anatomical structure of *Pyrgobaicalia* remains unstudied, and the author of the genus himself only provisionally classified it as a member of Baicaliidae [82].

As shown above, Baicaliidae diverged from their common ancestor with Amnicolidae during the Miocene–Pliocene, whereas the age of the recent Baicaliidae species flock is relatively young (3.5 Myr). The discrepancy is explained by the rapid speciation of a small number of species that passed through the genetic "bottleneck" [76].

The extinction of the majority of Miocene Baikal species (not only the baicaliids but also other mollusks and other invertebrates) occurred in the Upper Pliocene due to the global glaciation, tectonic rearrangements of the Baikal rift zone (including the formation of a single lake), and significant fluctuations in the water level [84–86]. The recent Baicaliidae are the descendants of a few ancestral species that survived this extinction event. The undoubted representatives of the ancestral baicaliids were found in the deposits of the Tankhoy formation (Oligocene-Miocene of the Central Baikal rift zone) [66,67]. We think that the taxonomic name Baicaliidae must be applied exclusively to the group of taxa from the Baikal rift zone. The attribution of the genus *Probaicalia* Martinson, 1949 (see Figure 9D) to the family Baicaliidae is questionable, despite the fact that its representatives have been found in the sediments of the northern Baikal rift zone (Vitim Plateau). The lake sediments of the Vitim Plateau were dated as Lower Cretaceous [66], and this geological age does not correspond to the estimated time of divergence of the Baicaliidae from a common ancestor with the Amnicolidae. However, we can assume that during the Cretaceous, this common ancestor was distributed in the ancient lakes of the Vitim River basin, where snails of *Kolhymamnicola* live now.

A number of other fossil species, shells of which are similar to shells of recent Baicaliidae, were found in the Late Cretaceous deposits of the Zabaikalie, Eastern Siberia, South-eastern Mongolia, Northern China, and Neogene deposits of Western Siberia and the Altay Republic [66,67,87–93]. These fossils belong either to the proamnicolid stem group or to gastropod families not closely related to the Amnicolidae sensu lato. In the latter case, their conchological resemblance to baicaliid snails is of convergent origin. Indeed, the high-turreted and sculptured shell, characteristic of the Baicaliidae, is known in representatives of other families, for example, in the Ponto-Caspian Pyrgulidae. Note, baicaliids, according to some authors [38,94], are a subfamily of the family Micromelaniidae (=Pyrgulidae). The fossil record of shells, which are conchologically similar to the recent Baicaliidae and, thus, classified within this family, is rather rich.

Representatives of eight species of the genus *Aenigmopyrgus* Popova and Starobogatov, 1970, conditionally assigned to Baicaliidae, were found in Neogene deposits of the Chuya and Kuray depressions of the Altai Mountains (Altai Republic, Russia) [92]. Their shells are well-preserved, which allowed Riedel et al. [95] to show the similarity of microsculpture of the protoconch of *Ae. steklovi* Popova and Starobogatov, 1970 to that of *Pyrgula annulata* (Linnaeus, 1767) from Lake Garda (northern Italy). The authors thus doubted the placement of *Aenigmopyrgus* in the family Baicaliidae and proposed to place this genus in Pyrgulidae.

Two species of *Palaeobaicalia* Martinson, 1961 were described from the Upper Cretaceous deposits of Western Mongolia, and, in our opinion, they are not related to the Baicaliidae, despite the conchological similarity (see Figure 9C). Their absolute geologic age predates the origin of the Baicaliidae substantially, and we propose here a new family,

Palaeobaicaliidae Sitnikova and Vinarski, **fam. nov**., with *Palaeobaicalia* as its type genus (Zoobank registration number: 58DE091D-2A3D-4D2B-8FF3-8E79F481ED93).

The diagnosis of the family is as follows. Shell small, up to 12 mm in height, fragile, its shape varies from subulate to high-conical and contains 7–10 weakly inflated or almost flat whorls. Fossil range—late Cretaceous–Paleogene.

We also include the genera *Gypsobia* Tausch, 1886 (Cretaceous of Hungary and East China) and *Probaicalia* (mentioned above) as well as two nominal extinct species of "*Baicalia*" described from China, Mongolia, and Western Siberia [66,88,90,96] to this new family. *Micromelania bicarinata* Martinson and Velikzhanina, 1960 (Lower Cretaceous of Western and Eastern Siberia and Mongolia) is probably another member of Palaeobaicaliidae, but its generic position is to be clarified.

The close relationship between Palaeobaicaliidae **fam. nov.** and the rest of the amnicolids can be doubted, as the shells alone do not give sufficient phylogenetic signals allowing establishing the phylogenetic position firmly. The findings of species of this group originate from a vast territory, ranging from Hungary in the west to Eastern China and Mongolia in the east, which corresponds to the broad distribution of the assumed ancestor of recent Amnicolidae and Baicaliidae (see above).

**Supplementary Materials:** The following supporting information can be downloaded at https://www.mdpi.com/article/10.3390/d15040483/s1: Table S1. List of the primers used in this study [97,98]; and Table S2 List of the species used for molecular analysis with accession numbers in GenBank.

**Author Contributions:** Conceptualization T.S., M.V. and L.P.; resources and data curations L.P., M.V. and E.B.; methodology T.S., T.P., D.S. and M.V.; visualization T.S. and T.P.; writing and revision T.S., M.V., T.P. and D.S. All authors have read and agreed to the published version of the manuscript.

**Funding:** M.V. and E.B. thank the Russian Scientific Fund for financial support of their research under the framework of Project No. 19-14-00066/P; investigations of T.S., D.S. and T.P. were supported by the state-funded projects conducted by LIN SB RAS Nos. 0279-2021-0007 and 0279-2021-0010.

**Institutional Review Board Statement:** Not applicable.

**Informed Consent Statement:** Not applicable.

**Data Availability Statement:** The primary data and materials for this study are placed in some public repositories (zoological museums); the newly obtained DNA sequences were submitted to GenBank. See the text for more information.

**Acknowledgments:** We are grateful to P. Kijashko and L. Yarokhnovich for their help in working with the ZIN RAS collection, A. Vorobieva for the preparation of histological slides to investigate female organs of *Kolhymamnicola*; M. Maslennikova and V. Egorov for their assistance with SEM equipment of the Collective Instrumental Centre 'Ultra microanalyses' at LIN SB RAS. We are thankful to the National Science Museum, Tokyo (Japan), and the Russian Geological Research Institute (VSEGEI, St. Petersburg, Russia) for the opportunity to examine their collections and take photos of gastropod shells. Our gratitude is extended to the anonymous reviewers of this work, for their useful comments and recommendations.

**Conflicts of Interest:** The authors declare no conflict of interest.

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
