# Peer review of "The North Asian Genus Kolhymamnicola Starobogatov and Budnikova 1976 (Gastropoda: Amnicolidae), Its Extended Diagnosis, Distribution, and Taxonomic Relationships†"

_diversity, doi:10.3390/d15040483_

Round 1

Reviewer 1 Report

The manuscript entitled „The North Asian genus Kolhymamnicola … , and taxonomic relationships“ is written in order to evidence resolving the taxonomic position and phylogenetic relationships of the endemic North Asian genus Kolhymamnicola. The evidence includes morpho-anatomical characteristics and a molecular phylogenetic analysis.

The manuscript is written in a very clear and detailed manner, including details about holotypes, paratypes, geographic localities, detailed morpho-anatomical analysis using light and electron microscopic images, fossil records, molecular data including primer sequences, publicly available COI, 16S and 18S sequences, carefully prepared unambiguous phylogenetic tree and sequence analyses performed using a whole lot of analytical tools.

I absolutely recommend publishing the manuscript and propose to check/correct the following:

line 168: „inlet of Lake Lama inlet“

line 210: „Table middle“

line 248: „has position“    and    „departs“

line 268: „bofy“

in Table 4, row Shell height: „up 7 mm“

line 303: „prises“

line 323: delete: „also were taken into account“

line 330-332 contains description of Table 5:    Together with it's description, Table 5 has to be moved and placed between lines 335 and 336.

Figure 9: I don't see scale bars (referred to in line 466)

line 537: replace „lake“ with „late“

line 549: „(?)“

line 563: „Paleogene (? Oligocene)“

Author Response

We are grateful to the reviewer for a positive evaluation of our manuscript and for pointing out errors in the text. We corrected most of them; please find below our detailed answer to the review:

line 168: „inlet of Lake Yama inlet“

corrected: an inlet of Lake Yama

line 210: „Table middle“

deleted

line 248: „has position“    and    „departs“

the seminal receptacle has rs2 position, and departs from

line 268: „bofy“

corrected - body

in Table 4, row Shell height: „up 7 mm“

corrected “up to 7 mm”

line 303: „prises“

Corrected: The second clade comprises the Far East genus Akiyoshia and the Southern Asian Moria and Erhaia.

line 323: delete: „also were taken into account“

These words have been deleted

line 330-332 contains description of Table 5:    Together with it's description, Table 5 has to be moved and placed between lines 335 and 336.

Yes, thank you, we have moved

Figure 9: I don't see scale bars (referred to in line 466)

Thanks, we add the scales to Figure 9

line 537: replace „lake“ with „late“

corrected

line 549: „(?)“

deleted

line 563: „Paleogene (? Oligocene)“

“(? Oligocene)” deleted

Reviewer 2 Report

The manuscript by Sitnikova et al. examines the gastropod genus Kolhymamnicola Starobogatov et Budnikova, 1976  of the family Amnicolidae. The study aimed to 1)  examine morphology of 4 currently valid species of the genus in order to provide a better diagnosis of the genus 2) determine the phylogenetic position of the genus in the subfamily Amnicolinae 3) examine ranges of the species within the genus. The authors examined specimens of all 4 species deposited in 2 (?) museums (unclear what abbreviations LIN and LMBI stand for) and collected additional specimens of K. wasiliewae, 8 of which were sequenced (but only 2 are later shown on the tree).

The results were 1) detailed morphological descriptions of 4 species based on the above material (perfectly in line with the aims) 2) literature-based comparisons of Kolhymamnicola species with those from 5 other genera of the family  Amnicolidae (Mastoniopsis, Taylorconcha, Amnicola, Akiyoshia, and Erchaia) and an unknown number of taxa of the family Baicaliidae (why this was done is not explained in the intended aims and was not mentioned in the M&M section) 3) a small phylogenetic tree (GenBank sequences of 10 specimens belonging to 9 species, including two newly obtained sequences of Kolhymamnicola wasiliewae, and an unknown number of sequences marked only as "Baicaliidae). 4) a map of Kolhymamnicola spp. records.

The phylogenetic tree clearly shows that the a) Kolhymamnicola forms reasonably supported sister group with Taylorconcha b) the family Amnicolidae is NOT monophyletic. What was used as outgroups in the phylogenetic analysis remains unknown.

The discussion states first that species composition of the genus did not change as a result of this study because of insufficient material, overlapping morphological characters and overlapping ranges, which is totally fine. However, the next section of discussion has nothing to do with the rest of the manuscript (material, methods or results) because it unexpectedly discusses fossil amnicolids! This section is purely speculative and should be deleted from the paper. The next section of the discussion suddenly proposes drastic nomenclature changes based on the very preliminary molecular tree - the authors propose creating a new family Erhaiidae (no diagnosis is given) for genera Erhaia, Akiyoshia, and Moria. This suggestion is clearly premature and is not justified by the data presented in the manuscript. In the final section of the discussion the authors propose yet another new family, this time a fossil one, Palaeobaicaliidae. Although a diagnosis was given, what this has to do with the results of this manuscript remains unclear. In summary, the conclusions are not only unsupported by the results, but are simply disconnected from the paper. The paper has to be re-written. I provide specific comments in the body of the manuscript attached.

Author Response

We are grateful for your critical remarks and editing the text. In most cases, we are agreeing with your recommendations and accept them with gratitude. However, in some points we are inclined to retain our original text and conclusions. The explanations for such decision are given below.  

Introduction

Lines 57-58 “The first nominal species of this genus, Amnicola miyadii Habe, 1942, was discovered in…”

Reviewer: So, this is a type species of the genus?

The type species of the discussed genus, Amnicola kolhymensis, was chosen in the original description. The species A. miyadii is the earliest described member of this genus but not the type one.

We have extended the last paragraph of the introduction to show the importance of discussing fossil shells in understanding the origin of the genus Kolhymanicola:

“The new molecular data, obtained during this study, allowed us to propose some changes to the current system of the Amnicolidae s. lato….”. “The first estimates of the age of evolutionary divergence of Kolhymamnicola and some other amnicolid taxa were obtained in this research. It allowed us to discuss the fossil records to understanding the origin Kolhymamnicola genus, and to reconsider the taxonomic status of some extinct genera, previously assigned to the family Baicaliidae, and to reject their close affinity to the living species of this group.”

Lines 101-102. The authors examined specimens of all 4 species deposited in 2 (?) museums (unclear what abbreviations LIN and LMBI stand for) and collected additional specimens of K. wasiliewae, 8 of which were sequenced (but only 2 are later shown on the tree).

Thanks. We have added, and corrected the Table 1:

Table 1. List of the studied material, including samples collected by the authors during this work, and kept in the Limnological Institute SB RAS, Irkutsk (LIN) and in the Laboratory of Macroecology & Biogeography of Invertebrates, St. Petersburg State University (LMBI).

And explained:

Sequences of all 8 specimens were registered in GenBank, only two unique haplotypes were revealed, and so these 2 haplotypes are shown on the tree.

M&M

We have added the information on the examined fossils:

Prior to dissections, the shells were photographed using a Canon EOS 60D camera with a Canon MP-E 65 mm f/2.8 1–5x macro lens. The same digital camera was used to take the photographs of the type specimens of the fossil gastropod species housed in the collection of the Russian Geological Research Institute (VSEGEI, St.-Petersburg, Russia). We examined specimens of 28 fossil species identified by G. Martinson as belonging to the families Hydrobiidae, Bithyniidae, and Baicaliidae sensu lato. These specimens were collected in Northern Asia, Baikal rift zone, Mongolia and China. Their fossil range varies from the Late Cretaceous to the Paleogene.

Line 113 Prior to investigation, the objects (replace to “specimens”) were rinsed in chlorine bleach….

The objects (protoconch, radulae, operculum etc) are listed in the first sentence of this paragraph; these parts (or objects) were sometimes belonged to a single specimen.

Line 117: Shells were measured according the standard scheme (technique?) [5], the measurements were performed on photographs using the ImageProPlus for Windows.

There are some scheme of the shell measured (Burch, 1988; Andreeva et al., 2010; Sitnikova, 2018 etc), we used the standard scheme (Starobogatov et al., 2004) similar to the scheme used by Starobogatov & Streletzkaja (1967) and Bogatov, Zatrawkin (1990), and we used the technique of the ImageProPlus for Windows, which does not need to be quoted by its authors.

Line 119 Molecular analysis was performed using 8 juvenile individuals of K. wasiliewae. How they are in Table?

The information is presented: K. wasiliewae, the second sample (Figure 8, locality 18), in the 3 column: 8 juveniles (used in the molecular analysis)

We added a sentence explaining the terminology of which authors we used when describing morpho-anatomical characters:

“In morpho-anatomical descriptions, we followed the terminology used in works of Radoman [30], Szarowska [31] and Hershler [32-35]”.

M&M molecular analysis

We corrected all errors in the text (underlined):

Gene fragments of the mitochondrial cytochrome c oxidase subunit I (COI), mitochondrial large rRNA (16S), and nuclear small rRNA (18S) were amplified in PCR using the primers listed in Table S1. From 1 to 3 μL of purified DNA was amplified in a 25 μL reaction mixture using the BioMaster HS-Taq PCR Kit (Biolabmix, Russia) following the manufacturer’s recommendation. We amplified all gene fragments using a temperature profile of 94°Ð¡ 4 min (94 °Ð¡ 1 min, 50 °Ð¡ (55 °Ð¡ for 18S rRNA) 1 min, 72 °Ð¡ 1 min) × 30 cycles, 72 °Ð¡ 5 min. The amplicons were visualized on 1% agarose gel. Sequencing was performed using an ABI 3130 automated sequencer (Research and Production Company "SYNTOL," Moscow, Russia). The nucleotide sequences were verified manually, and aligned using default settings in CLUSTAL W [42], implemented in BIOEDIT v.7.2.5 [43]. The resulting COI nucleotide alignment was translated to amine acids sequences to make sure that stop codons are absent. 16S and 18S nucleotide sequences were aligned with MAFFT v. 6.2 (e-ins-I algorithm) [44]. The obtained nucleotide sequences were compared by BLAST with those for species belonging to 6 genera of the Amnicolidae and 19 species of 8 genera of the Baicaliidae accessible from GenBank. Accession numbers of the new and retrieved nucleotide sequences presented in Table S2. Maximum Likelihood (ML) phylogeny was inferred using IQ-TREE v.1.6.8 and midpoint rooted [45]. The most suitable model of molecular evolution was chosen using the Model Finder module within IQ-TREE [46]. Branch support was assessed using bootstrap values [47] and the Shimodaira–Hasegawa approximate likelihood ratio test (SHaLRT; see [48]). The time of divergence of K. wasiliewae from the most recent common ancestor (tMRCA) was estimated using the COI sequence dataset by a Bayesian approach with BEAST 1.10.4 [49]. The analysis was performed using a GTR+I+G model of nucleotide substitution selected using JMODELTEST. The suggested divergence rate was set to 1.96% per Myr, following Wilke et al. [50]. The Birth–Death model [51] was applied to a speciation process for our data set. The analysis was run for 100,000,000 generations, with the first 10,000,000 discarded as burn-in and parameter values sampled every 10,000 generations. The results were analyzed with TRACER 1.6 [52] to assess the convergence and confirm that the combined effective sample sizes for all parameters were larger than 200, ensuring that the Markov Chain Monte–Carlo (MCMC) had run long enough.

[51] The reference was added: Stadler, T., Kouyos, R., von Wyl, V., Yerly, S., Böni, J., Bürgisser, P., ... & Swiss HIV Cohort Study. (2012). Estimating the basic reproductive number from viral sequence data. Molecular biology and evolution, 29(1), 347-357)

Results

We have corrected all the text according to your recommendations, including insertion of the Museum abbreviations and accession numbers of the examined specimens presented in the Figures; corrected Table 2, replaced Figure 8; corrected Discussion and we made an extensive linguistic polishing of the original text.

What was used as outgroups in the phylogenetic analysis remains unknown.

The phylogenetic tree was midpoint rooted. Midpoint rooting calculates tip to tip distances and then places the root halfway between the two longest tips (Swofford et al., 1996; Gladstone, Whelan, 2022).

The next section of discussion has nothing to do with the rest of the manuscript (material, methods or results) because it unexpectedly discusses fossil amnicolids! This section is purely speculative and should be deleted from the paper. The next section of the discussion suddenly proposes drastic nomenclature changes based on the very preliminary molecular tree - the authors propose creating a new family Erhaiidae (no diagnosis is given) for genera Erhaia, Akiyoshia, and Moria. This suggestion is clearly premature and is not justified by the data presented in the manuscript. In the final section of the discussion the authors propose yet another new family, this time a fossil one, Palaeobaicaliidae. Although a diagnosis was given, what this has to do with the results of this manuscript remains unclear. In summary, the conclusions are not only unsupported by the results, but are simply disconnected from the paper. The paper has to be re-written.

We explained in Introduction the importance of this discussion, and we add the text about examined fossils in M&M.

We are not describing the family Erhaiidae as a new one. All what we propose is to elevate the rank of the previously described tribe Erhaiini to the rank of family. According to ICZN, no diagnosis is needed in this case, since the diagnosis of this taxon was published in its original description (Davis & Kuo, 1985). Table 4 of our manuscript provides a set of morphoanatomical traits allowing one to distinguish members of Erhaiidae and other amnicolid genera. The rationale benind the elevation of taxonomic rank of Erhaiini is explained in the text of our work. If we treat Baicaliidae as a family of its own, the original rank of Erhaiini cannot be retained since this group appeared to be more distant from the Amnicolidae (s. str.) than the clade combining the living baicaliid snails. Retaining Erhaiini within Amnicolidae would make the latter taxon paraphyletic. To elevate the rank of Erhaiini is the only available decision in this case.

The part concerning fossil taxa formerly treated as belonging to Amnicolidae s. lato and/or Baicaliidae is needed, and its content is directly related to the part discussing the living amnicolids. The results of our work provide a molecular estimate of the absolute age of divergence of the family Baicaliidae, which took place in Miocene (or, possibly, in Late Oligocene). The currently accepted system of this family includes a series of genera of much older age (see: https://www.molluscabase.org/aphia.php?p=taxdetails&id=819717). These genera cannot belong to the clade containing living Baicaliinae since they lived several decades of million years earlier! It would be unscientific either to leave them within this family or to let them to dwell in a “no man’s land” as belonging to no family. In this situation, we decided to separate this group of genera in a special family. It is the only way to avoid discrepancy between the fossil record and the molecular phylogeny. We are not think that this paleontological part is purely speculative. We were able to examine these fossil remains and provide the photos of some of them. Another reason to publish these data in this paper is that the fossil material is rather limited and its volume is not large enough to prepare a separate publication.

Round 2

Reviewer 2 Report

In my opinion the authors successfully addressed the comments and revised the manuscript accordingly